# The latest improvements with SURFEX v8.0 of the Safran-Isba-Modcou hydrometeorological model for France

Patrick Le Moigne[1], François Besson[2], Eric Martin[1,3], Julien Boé[4], Aaron Boone[1], Bertrand Decharme[1], Pierre Etchevers[2], Stéphanie Faroux[1], Florence Habets[5], Matthieu Lafaysse[6], Delphine Leroux[1], Fabienne Rousset-Regimbeau[2]

[1] CNRM, Université de Toulouse, Météo-France, CNRS, Toulouse, France
[2] Direction de la Climatologie et des Services Climatiques, Météo-France, Toulouse, France
[3] Irstea, Université d'Aix Marseille, RECOVER, Aix-en-Provence, France
[4] CECI, Université de Toulouse, CERFACS, CNRS, Toulouse, France
[5] CNRS UMR 8538, Laboratoire de Géologie, Ecole Normale Supérieure, PSL Research University, 24 rue Lhomond, 75005 Paris, France
[6] CNRM, Université de Grenoble, Météo-France, CNRS, Grenoble, France

*Correspondence to*: Patrick Le Moigne (patrick.lemoigne@meteo.fr)

**Abstract.** This paper describes the impact of the various changes made on the Safran-Isba-Modcou hydrometeorological system (SIM), and demonstrates that the new version of the model performs better than the previous one by making comparisons with observations of daily river flows and snow depths. SIM was developed and put into operational service at Météo-France in the early 2000s. The SIM application is dedicated to the monitoring of water resources, and can therefore help in drought monitoring or flood risk forecasting on French territory. This complex system combines three models: SAFRAN which analyses meteorological variables close to the surface, ISBA land surface model which aims to calculate surface fluxes at the interface with the atmosphere and ground variables, and finally MODCOU a hydrogeological model which calculates river flows and changes in groundwater levels. The SIM model has been improved first by reducing the infrared radiation bias of SAFRAN, and then by using the more advanced ISBA multi-layer surface diffusion scheme to have a more physical representation of surface and ground processes. In addition, more accurate and recent databases of vegetation, soil texture and orography were used. Finally, in mountainous areas, a sub-grid orography representation using elevation bands was adopted, as well as the possibility of adding a reservoir to represent the effect of aquifers in mountainous areas. The numerical simulations carried out with the SIM model covered the period from 1958 to 2018, thereby providing an extensive historical analysis of the water resources over France.

## 1 Introduction

The coupling of hydrogeological models and land surface models (LSMs) aims to represent the water cycle by considering as many physical processes as possible. Thus, in LSMs, precipitation that reaches the ground contributes to water storage, evaporation, surface runoff and infiltration into the soil. In addition to the water balance, LSMs simulate the surface energy balance, which is closely related to the water balance in terms of evaporation. In such a coupled system, surface runoff is

collected by the surface river system while deep infiltration of the soil contributes to aquifer recharge. Such systems have been used for decades to study water resources and predict their evolution. Land surface models (LSMs), whether or not coupled to hydrological models, have been the subject of numerous studies that have improved them over time and have led to a better description and understanding of the key processes governing exchanges at the interface between the surface and the atmosphere and the surface and the subsurface. These studies, which include international measurement campaigns or more regional an even local initiatives, have made it possible to evaluate surface models, and even certain parameterizations, by comparing simulation results with different types of observations such as in situ measurements, reanalyses or satellite products. Simulations were carried out offline, i.e. decoupled from the atmosphere, to limit the impact of potential atmospheric biases in the surface schemes, by constraining atmospheric forcing through observations, when possible. The first international model intercomparison projects were the Project of Intercomparison of Land surface Parametrization Schemes (PILPS), described in Henderson-Sellers et al. (1996), which began with forcing from atmospheric simulations (Pitman et al., 1993) and, in a second stage, forcing from local observations (Chen et al., 1997). The successive phases also focused on different issues, such as snow and frost parameterization (Schlosser et al., 2000), river flow assessment (Wood et al., 1998; Bowling et al., 2003). In the spirit of PILPS, GSWP (Global Soil Wetness Project, Dirmeyer, 2011) was initiated with global scale simulations. The results of this project are the first global offline multi-model simulations of LSMs. Other more specific intercomparison projects have been carried out such as SnowMIP (Etchevers et al., 2004) to study snow-related processes, ALMIP (Boone et al., 2009), focusing on critical surface processes in West Africa at regional scale, or Rhône-AGGrégation (Boone et al., 2004) to study coupling with hydrology. More recently, the PLUMBER project (Best et al., 2015) has attempted to identify how LSMs behave in relation to certain benchmarks and to define performance criteria that LSMs should be able to achieve according to the information available in atmospheric forcing, thus avoiding direct comparison with observations.

In many of these intercomparison studies, the surface models were validated at the local scale and used average parameters that were known fairly accurately. However, these models sometimes have strongly non-linear components, such as the link between root zone moisture and transpiration when the soil dries out (Sellers et al., 1997), so it is necessary to develop sub-grid parameterizations to compensate for the lack of representativeness of the mean parameters. Overgaard et al. (2006) conducted a review of surface models based on energy balance and used for hydrological purposes. They stressed the need to validate the models at the local scale, but also showed the interest of using remote sensing data to evaluate the models. Indeed, the validation of LSMs using river flows alone does not prove that surface fluxes, for example, are well simulated by the model and that there is no error compensation. Furthermore, estimating surface fluxes by remote sensing is not straightforward and requires certain assumptions that are not always valid, and inversion models are used to translate the remote sensing measurement into a model variable equivalent. However, using surface fluxes to validate surface models is also subject to questioning since the energy balance measured at the surface is generally not closed (Foken, 2008) whereas it is an imposed constraint in surface models. The use of international measurement networks such as FLUXNET (El Mayaar et al., 2008; Napoly et al., 2017) is also widely used to evaluate surface models at the point scale. Remote sensing provides a

means of observing hydrological state variables over large areas (Schmugge et al., 2002) and can be useful in the case of LSMs coupled to hydrological models, in particular in order to assess evaporation (Kalma et al., 2008; Long et al., 2014; Wang et al., 2015) or soil moisture (Goward et al., 2000; Albergel et al., 2012; Fang et al., 2016). It should be noted that these remote sensing data can be assimilated to correct the model state variables at the initial time as well as during the hindcast (Albergel, et al., 2017).

In addition, climate models have been evaluated at both global and regional scales through hydrology. Indeed, the coupling between their land surface model and hydrology allows a quantitative assessment to be made, through comparisons to variables such as river flow, groundwater levels and snow depth. This is the case of river flows simulated by hydrogeological models, which can be compared with in situ measurements from gauging stations (Habets et al., 2008; Decharme et al., 2013; Alkama et al., 2010; Barthel and Banzhaf, 2015; Decharme et al. 2019). The coupling between LSMs and hydrogeological models in water resource studies is an appropriate tool for answering scientific questions such as the importance of climate change on these resources (Vidal et al., 2010; Dayon et al., 2018; Bonnet et al., 2018) or how human activity influences these resources (Martin et al., 2016; Biancamaria et al., 2019). Recent initiatives to study the impact of anthropization on water availability, such as those supported by the Global Energy and Water Exchanges (GEWEX) project (Harding et al., 2015), where the contribution of LSMs to modelling appears to be important, show that irrigation needs to be considered in the models (Boone et al., 2019).

At Météo-France, the SIM system was first designed to study the water cycle in major French basins such as the Rhône basin (Etchevers et al., 2000), the Adour basin (Habets et al., 1998), the Garonne basin (Voirin et al., 2002) and finally all of France (Habets et al., 2008). This system has been shown to be very useful for many applications. For example, since 2003, the SIM system has been used operationally at Météo-France for drought monitoring, this is done using hindcast simulations, in addition to near real-time applications. These applications in France were based on an LSM using the Force-Restore approach (Noilhan and Planton, 1989; Noilhan and Mahfouf, 1996) for heat and water transfer in the soil. However, this method has some limitations in terms of the realism of certain physical parameterizations, which are detailed in section 2.2. These limitations concern the representation of snow, the interactions between snow and ground freezing which are not always well represented, the description of the vertical profile of roots in the soil, or the composite approach to represent vegetation, mixing different types of vegetation into one, with aggregated characteristics. Although this method has proven, over the last decades, to be suitable for addressing scientific issues related to water resources, a more physical approach, based on the diffusion of heat and moisture in the soil (Decharme et al., 2011), has been developed to consider more sophisticated numerical schemes and improve system performance.

The objective of this paper is to show how the development of new parameterizations and better atmospheric forcing prescription have improved the performance of the system. The current study, based on numerical simulations covering the period 1958-2018, shows how improvements in atmospheric forcing, land surface model physics and subgrid orography and hydrology improve the modelled river flow and snow depth of the SIM system. It also aims to describe how the model results are affected by each change separately and finally to demonstrate that the new model configuration performs better

than the previous one in terms of river flow extremes, and when simulated snow depth or average river flow is compared to observed data.

Section 2 describes the original SIM configuration and its recent updates. Section 3 presents climate data and evaluation datasets, and the offline experiments used to demonstrate the advantages of the new SIM system. In section 4 the results of the new system are presented and finally they are discussed in the last section.

## 2. SIM hydrometeorological model

### 2.1 Overview of the 2008 version of the model

The SIM hydrometeorological model (Habets et al., 2008) combines the three models SAFRAN, ISBA and MODCOU. SAFRAN (Durand et al., 1993; Quintana Segui et al., 2008) performs a 6-hourly analysis of near-surface meteorological variables such as temperature and relative humidity at 2 metres, wind speed, cloud cover and a daily analysis of 24-hour accumulated precipitation. The analysis is carried out over geographical areas covering a few hundred square kilometres (Le Moigne, 2002), and the analysed fields are interpolated to hourly time steps. Direct and diffuse solar radiation and infrared

radiation are calculated from the analysis of cloud, temperature and humidity profiles using a radiative transfer model (Ritter and Geleyn, 1992). A spatial interpolation is then performed on a regular horizontal 8 km grid to provide the ISBA land surface model (Noilhan and Planton, 1989; Noilhan and Mahfouf, 1996) with the necessary climate information. The grid is composed of 9892 cells (Fig. 1, left panel) covering France and is extended beyond the borders to include the upstream part of the catchment basins.

The ISBA model uses SAFRAN analysis as input and calculates the surface energy and water budgets over the vegetated areas. The water budget in ISBA ensures that soil moisture results from the balance between water input from incoming precipitation and water losses due to surface evaporation, surface runoff and infiltration into the soil. These last two components are fed into the hydrogeological model MODCOU (Ledoux et al., 1989; Habets et al., 1998) in order to calculate the temporal evolution of river flows for a given set of gauging stations and groundwater head where aquifers are

simulated, i.e. on the Seine and Rhône basins only (delimited by the yellow zones in Fig. 1, right panel). The original SIM system differs in many respects from the version described in this document.

### 2.2 Improvements to the land surface model in SURFEX v8.0

In the original SIM system, heat and water transfers in the soil were based on the Force-Restore method (Noilhan and Planton, 1989; Noilhan and Mahfouf, 1996, Decharme et al., 2011) which has been widely used in research for decades and

130 is still operationally used in the French global numerical weather prediction model ARPEGE (Courtier and Geleyn, 1988) and the mesoscale model AROME (Seity et al., 2011). In the Force-Restore method, the soil is divided into two layers for

temperature and three layers for moisture (Boone et al., 1999). However, such a method has shown some limitations in the representation of surface and soil processes such as the interaction between snow and soil freezing (Luo et al., 1998) due to vertical discretization or the inability to correctly represent the vertical profile of roots in the soil (Braud et al., 2005) and thus the vertical transfers of moisture and heat. The alternative of using the Force-Restore method was developed by Boone et al. (1999) and revisited by Decharme et al. (2011) who proposed to use diffusive equations to solve both heat and water transfer equations in the soil, based on Fourier and Darcy laws, respectively. Such a method proposes a discretization of the soil into 14 layers, resulting in a total depth of 12 meters, with a fine description of the subsurface layers to capture the diurnal cycle. The vertical discretization (bottom depth of each layer in metres) is as follows: 0.01 m, 0.04 m, 0.1 m, 0.2 m, 0.4 m, 0.6 m, 0.8 m, 1 m, 1.5 m, 2 m, 3 m, 5 m, 8 m, 12 m, as described in Decharme et al. (2013). Heat transfer is resolved over the total depth, while moisture transfer is resolved only over the depth of the roots, which depends on the type of vegetation and its geographical location: a maximum of 1.5 m for type C3 crops and 2.5 m for forests in France. In such a model, soil temperatures and soil moisture are calculated at the same nodes, which is necessary to correctly represent soil freezing for example. Another notable improvement concerns snow modelling. The original three-layer snow scheme developed by Boone and Etchevers (2001) aimed to represent the physical processes in the snow realistically with a simple model, and for this purpose some processes had been adapted from the Crocus snow model (Vionnet et al., 2012) for snow avalanche forecasting. The main new features recently developed and introduced into the ISBA snow model are described in detail in Decharme et al. (2016) and concern (i) snow stratification with an increase in the number of layers close to the surface in contact with the air, but also with the ground to better represent the diurnal cycle and heat transfer at the interface with air and ground, respectively, (ii) snow compaction due to changes in viscosity (Brun et al., 1989) and wind-driven densification at the surface (Brun et al, 1997), and (iii) snow absorption of solar radiation as a function of 3-band spectral albedo.

Second, the representation of vegetation in the model has also evolved from the original version, where vegetation types within a grid cell were aggregated with averaged surface parameters (Noilhan and Lacarrere, 1995), whereas the new system uses 12 separate vegetation types, each with its own set of parameters (Masson et al., 2003; Faroux et al., 2013). The classification distinguishes three non-vegetated types: rocks, bare soil and permanent snow and ice, and nine vegetated types: temperate deciduous forest, boreal conifers, tropical conifers, C3 crops, C4 crops, irrigated crops, grasslands, tropical meadows, and peatlands, parks and gardens. Although this approach is more computationally intensive time because the model must be run for each vegetation type, the realism of ISBA simulations is increased because the parameters better characterize the contrasting surface properties. In addition, the explicit use of 12 vegetation types is mandatory when using ISBA-A-gs, the simplified photosynthesis module of ISBA (Calvet et al., 1998) aimed at representing a realistic photosynthesis of the different biomes. In the new version, the drought avoidance or drought tolerance response is adopted (Calvet et al., 2004).

Hydrological processes are obviously important in a system for calculating the water budget of natural surfaces and simulating river flows. The old parameterization of drainage, developed by Mahfouf and Noilhan (1996) for the Force-Restore scheme, has been replaced by a method of diffusing water into the soil. In ISBA, surface runoff occurs over saturated areas (Dunne and Black, 1970). Habets et al. (1998) proposed sub-grid parameterization to generate surface runoff over grids of several square kilometres before the entire grid is saturated, in order to consider some regional heterogeneities in infiltration arising from orographic variability or precipitation spatial inhomogeneity. In this approach based on sub-grid variability of topography, and used in the VIC model (Liang et al., 1994; Dümenil and Todini, 1992), the fraction of the saturated zone varies as a function of the water content of the soil and a curvature term $b$ that must be calibrated. In the original system $b$ is equal to 0.5, a very high value compared to other studies at the watershed scale. Indeed, a more realistic value should be around 0.2 (Lohman et al., 1997; Ducharne et al., 1998). However, the Force-Restore scheme is known to be too dry in terms of soil moisture (Decharme et al., 2011, 2019) and a steep slope (therefore a fairly large curvature term) in the grid mesh is required to generate sufficient runoff in certain regions. The use of the diffusion scheme has removed this constraint of a high $b$-factor and in the new SIM application, a value of 0.25 is now used on zones without aquifers where it is set at 0.01 corresponding to the absence of sub-grid runoff. Dümenil and Todini (1992) have parameterized the fraction of saturated zone $A$ as a function of soil moisture $A(w_2) = 1 - \left(1 - \frac{w_2}{w_{sat}}\right)^{b/(b+1)}$, where $w_2$ is the volumetric water content of the soil in the rooting zone and $w_{sat}$ its value at saturation, and for a loamy zone ($w_{sat} = 0.45$ m$^3$/m$^3$) of wet soil ($w_2 = 0.4$ m$^3$/m$^3$), the presence of an aquifer ($b = 0.01$) is characterized by a small area of saturated fraction of about 2%.

## 2.3 Use of more precise parameters for the land surface

In addition to the changes in model physics described above, the land cover and topography databases have been updated to improve the realism of the external parameters of the ISBA model. The hydrogeological database representing the aquifer and the routing network was unchanged. In addition, the soil texture database for France is unchanged. In the former SIM system, the soil texture was based on a soil map provided by the Institut National de Recherches Agronomiques (INRA - King et al., 1995) at a resolution of 1 km. In the new SIM system, texture is defined by the Harmonized World Soil Database (HWSD - Nachtergaele et al., 2012) which is a soil map at 1 km resolution that combines several data sets available worldwide. In particular for France, the INRA soil map mentioned above has been integrated into the HWSD dataset (used in other applications using SURFEX outside France), so this change does not affect the SIM simulations.

The topography, derived from the 30 arc second global elevation data (GTOPO30, http://eros.usgs.gov/#/Find_Data/Products_and_Data_Available/gtopo30_info), has been replaced by that of the "Shuttle Radar Topography Mission" (SRTM90, https://cgiarcsi.community/data/srtm-90m-digital-elevation-database-v4-1/) at a 90 m resolution (Figure 1, left panel). Note that the impact of using SRTM90 is rather limited because the target grid resolution for SIM applications over France is 8 km, which implies that small scale differences between the orography data is averaged at such a resolution (thus the SIM topography is similar, whether GTOPO30 or SRTM90 is used).

The last modification of the input database is the vegetation map which provides the fraction of each ecosystem. The global 1 km resolution map ECOCLIMAP1 (Masson et al., 2002) was originally used in the SIM application for France. Subsequently, a new classification algorithm was developed over Europe, the ECOCLIMAP2 land-use map (Faroux et al., 2013), in order to use more accurate and recent satellite information as input and for a longer period. Among the differences

to note, ECOCLIMAP1 used, for example, AVHRR satellite data from 1992-1993 whereas ECOCLIMAP2 uses SPOT/VEGETATION data from 1999-2005. The impacts of modifying the vegetation fraction input to the ISBA model are multiple and will not be described here in detail (for a detailed comparison, see Faroux et al., 2013). ECOCLIMAP2 has definite advantages, the effects of which are directly reflected in the ISBA model. For example, ECOCLIMAP2 covers a larger time period than the previous version and therefore allows a better representation of the variability of surface

parameters. Also, it distinguishes different types of crops that can be modelled separately, and therefore more accurately, with ISBA. The sensors on board satellites have better accuracy and the uncertainty of the measurement is reduced. The vegetation fraction in particular is improved and with it the roughness length of the vegetation which impacts the surface wind by the obstacle effect on near-surface flows. The leaf area index is also improved and its increase leads to a better description of the evaporative fraction, which is key for the energy partitioning in the model. The more realistic surface

albedo developed by Carrer et al. (2014) was also used, as Decharme et al. (2013) showed that it improved results at the global scale.

### 2.4 Evolution of downward infrared radiation

SAFRAN radiation has been corrected to compensate for a radiation deficit already identified in several studies (Le Moigne et al., 2002; Carrer et al., 2012, Decharme et al., 2013) although observations of this variable are very rare. Radiation in

SAFRAN is simulated (Ritter and Geleyn, 1992) from an analysis of cloud cover based on analyses of temperature and humidity profiles from the French global atmospheric model ARPEGE. Le Moigne (2002) and Carrer et al. (2012) showed that SAFRAN's infrared radiation was weakly biased, and Decharme et al (2013) increased overall infrared radiation over France by 5% in their off-line simulations. The bias is likely due to a problem in the analysis and in the radiative transfer (RT) model. The cloud cover analysis is computed using temperature and humidity profiles from a large-scale atmospheric

model that contains biases. Moreover, the model used to solve the RT is an old model, with a rather low vertical resolution and therefore probably sub-optimal, but which was state-of-the-art in the 1990s. For example, Le Moigne (2002) showed that infrared and solar radiation were too low at the Col de Porte site in the Alps and proposed a correction for cloud cover and altitude which was successfully applied to the Rhône basin in the Rhône-AGG intercomparison experiment (Boone et al., 2004). In this study, only the infrared correction is considered and applied over the whole French territory. The infrared

correction, described in Appendix A, was established by comparing the SAFRAN analysis and the infrared measurements of two meteorological stations, Carpentras and Col de Porte, which are reference stations for infrared measurements monitored by Météo-France and located in the south-east of France and in the Alps, respectively. Carpentras is located in the plains while the Col de Porte, an experimental measurement site of the Centre d'Etudes de la Neige, is located in the French Alps at

an altitude of 1340 m (Morin et al., 2012; Lejeune et al., 2018). The correction is only applied below 1340 m, as a positive bias is found at the Saint-Sorlin site (Queno et al., 2017). Figure 2 shows the annual average over the 60-year period initial infrared radiation (left panel) and the amount of energy supplied when the correction is applied (right panel).

## 2.5 Altitudinal subgrid variability in mountainous areas

In SAFRAN, the analysis is performed on homogeneous zones of several hundred square kilometres and the vertical component is explicitly considered with to a 300-metre slicing along the vertical. For each grid cell $i$, the analysed variables $X^a(i)$ are then interpolated on an 8 km horizontal grid, considering the average altitude of each grid cell. The analysed variables are then used as input to the ISBA surface model. At this resolution, the 9892 grids cover all of France as well as some border areas for hydrological purposes. However, this resolution is still too coarse to accurately capture the variability of certain variables, particularly in the mountains. Lafaysse et al. (2011) demonstrated in the Durance basin that the use of altitude bands was an efficient method to better describe the spatial variability of the snow cover and its impacts on river flows, at a numerical cost much lower than increasing the horizontal resolution. A similar approach was therefore defined for the entire French territory and can be summarized as follows: for a given mesh $i$, the SAFRAN analysis is performed every 300 m and the $X^a(i, k)$ are the $k$ sets of analysed variables corresponding to each of the $k$ altitude bands. For each $i$, if the vertical sub-grid variability is sufficiently large, a complementary set of $k'$ elevation bands is defined for different elevations in order to represent this variability. Vertical interpolation is then performed on the atmospheric forcing at each $k'$ band. For each $k'$ band, ISBA simulates surface runoff and soil infiltration which are used to calculate the total surface runoff and soil infiltration for grid point $i$. Of the 9892 grid points, 1044 are above 500 m and have a high variability in sub-grid topography. Using a vertical discretization of 300 m at each grid point to represent topographic variability was ideal but too costly. A solution based on the distribution of elevations in each grid cell into five bands represented by the quintiles q20, q40, q60 and q80 was adopted. For each of the 1044 grid points, the vertical discretization varies spatially, and the vertical discretization is variable when the maximum altitude difference in the grid exceeds 300m. Thus, for all of the 1044 points, the minimum difference (23m) between two consecutive bands is obtained in medium mountains, for altitudes of 385m, 694m, 717m, while the maximum difference (986m) is obtained in high mountains for altitudes of 525m, 861m, 1847m, 2013m. In the end this gives a total of 3878 grid points involved in the calculations of the mountain simulation. Figure 1 (right panel) shows the elevation of the 1044 grid points where the elevation band method is applied.

In addition, to compensate for the inability of the SIM system to simulate low flows when aquifers are not explicitly considered, subgrid drainage parameterization was used in the original SIM system. This subgrid drainage is controlled by a parameter calibrated for both lowland and mountain areas, but such a calibration does not work very well because the water used to support low flows is taken from the rooting zone and not from the aquifer. In the new system, this parameterization is removed, and a parameterization has been added to mimic the behaviour of a deep reservoir to support low flows and to limit

peak flooding due to snowmelt (Lafaysse et al., 2011). Retaining water due to snowmelt and releasing it during the dry season made it possible to simulate peak flooding, but summer low flows are still underestimated.

## 3. Design of experiments and data sets

### 3.1 Offline simulations

The SIM system is an off-line application where the ISBA land surface model is driven by climate data and there is no
feedback from the surface to the atmosphere. Different SIM configurations were designed to highlight the improvements achieved, with each simulation being equilibrated using a two-year spinup.

The first configuration refers to the old SIM system (SIM_REF below, as described in 2.1), i.e. before any changes described above. An additional reference simulation (SIM_REF2 below) is based on SIM_REF where subgrid drainage is removed. The first experiment (SIM_PHY hereafter) consists in modifying the physics and input databases. SIM_PHY uses the
diffusion scheme with 14 layers in the soil, the improved snow scheme with 12 layers, a tile approach based on 12 vegetation types, and a runoff parameterization where the high constraint on the coefficient $b$ ($b$=0.5 in the runoff parameterization in SIM_REF) has been lowered to 0.25. Also, in SIM_PHY, updated databases are used for a better representation of soil texture, orography and vegetation. The correction of SAFRAN infrared radiation according to cloud cover is then introduced in the SIM_FRC experiment (based on SIM_PHY). Then SIM_TOP (based on SIM_FRC) uses the representation of subgrid
orography in mountains, and finally SIM_NEW (based on SIM_TOP) considers a drainage reservoir in mountains. Table 1 summarizes the main characteristics of the experiments.

In the SIM system, climatic data are provided by the SAFRAN analysis. In this study, SAFRAN covers a 60-year period, from August 1, 1958 to July 31, 2018. In SAFRAN, the guess of the analysis used is ERA-40 until 2002 and the ECMWF operational analysis thereafter. In France, the density of the observation network is very high, because a network dedicated
to climatology completes the less-dense synoptic network. There are therefore practically no regions with poor coverage, especially for precipitation, which is essential for hydrology, and the coarse resolution of the analysis first guess is not an issue. The analysed variables are then interpolated every hour on the SIM grid at a resolution of 8 km and this complete set of near-surface variables is then used to conduct off-line simulations. The averages of the fields analysed or reconstructed by SAFRAN over the entire period over France, used as input data for off-line experiments, are shown in Figure 3, while the
annual averages of these quantities are shown in Figure 4.

### 3.2 Data sets and validation tools

Various datasets were used to evaluate the performance of the SIM model throughout the validation process to ensure that an improvement in the input climate data or physics simultaneously improved the surface or ground variables and river flows.

The Land Surface Analysis Satellite Applications Facility (LSAF) project disseminates products based on data from the Meteosat second generation geostationary satellites, and, in particular, downwelling infrared radiation (LSA SAF; Trigo et al., 2011; http://lsa-saf.eumetsat.int ). LSAF data covering the period from August 1, 2010 to July 31, 2015 are used here to assess the quality of SAFRAN's infrared radiation.

In addition to the infrared radiation data from Carpentras and Col de Porte already mentioned in section 2.4, in situ data from
the French GLACIOCLIM observation service (https://glacioclim.osug.fr) stations at Saint-Sorlin (2620 m) and Argentière (1900 m) were also used to assess SAFRAN's infrared radiation at altitude. The period covered runs from December 2005 to December 2015.

River flow observations are taken from the French national database Banque Hydro (http://hydro.eaufrance.fr/) from 1958 to 2018. Daily and monthly flow data from 470 selected gauging stations were used to evaluate river flows simulated by the
MODCOU hydrogeological model. Only gauging stations with observations for at least half of the days over the total period were kept. The efficiency of Nash Sutcliff (NSE, Nash and Sutcliff, 1970) was used to evaluate the performance of the model, and in addition the flow ratio between SIM simulations and observations was calculated to assess the bias of the system. The complementary cumulative distribution function (CCDF, below) of the NSE, which calculates the probability that the NSE is greater than a threshold, averaged over the number of gauging stations in France, is also used as a measure to
evaluate the NSE.

Observed snow depth is another independent data set (i.e. not assimilated in the reanalysis process) used to evaluate the system. Measurements from 185 stations in the Alps, the Pyrenees and Corsica at altitudes between 600 and 3000 m above sea level are used. They include 26 ultrasonic sensors (located mainly in high-altitude areas: the Nivose network) and 161 stations operated by Météo-France partners, mainly in ski resorts which are manual measurements using snow sticks. The
daily total snow depth is used to calculate the bias and root mean square errors for the SIM_REF and SIM_NEW simulations over the period 1984-2016 between October and June. Note that most stations do not provide complete data for the entire period. The length of the measurement series and the number of seasons that stations are open are sources of variability in the scores. However, since very few series are complete, the choice was made to evaluate the performance of the model by considering as many stations as possible rather than trying to homogenize the length of the series.
The SAFRAN analysis is performed on homogeneous zones in terms of horizontal gradients, and the analysed fields are spatially interpolated to a regular 8 km grid taking altitude into account. Thus, the comparison of infrared radiation (IR) is made between the SAFRAN analysis interpolated at 8 km and the local observation. The horizontal variability of IR radiation at 8 km is small enough to allow a direct comparison with in situ observations. Moreover, the ISBA model outputs of ground temperature and snow depth profiles are relatively sparse and only a direct comparison between the model outputs
and the observations is possible. Finally, with respect to river flows, the MODCOU model grid varies in the range of 8 km to 1 km near the riverbed, and the comparison between the model output and the observed flow is made by considering the flow at the river outlet and the corresponding model grid point in the 1 km hydrological network grid. This way of locally

validating models by comparing the observation to the corresponding model grid point is not new and has been used in many studies in France and elsewhere (Habets et al., 2008; Decharme et al., 2013; Lafaysse et al., 2011; Vergnes et al., 2014).

## 4. Results

### 4.1 Description of climate data

Figure 4 shows the annual averages of atmospheric forcing from 1958 to 2018. The 2-metre air temperature (Fig. 4a) and specific humidity (Fig. 4b) show natural interannual variability and a tendency to increase over time by about 1.4 K and 0.6 g Kg$^{-1}$ (linear regression of the time series of annual means), respectively. The abrupt change in temperature in 1987/1988, referred to by Brulebois et al (2015), is not so obvious to explain. The 10-metre wind speed (Fig. 4c) at the beginning and end of the analysis period is of the same order with an amplitude of 2.8 m s$^{-1}$, but with a significant decrease of 0.5 m s$^{-1}$ between 1983 and 1995, followed by a steady increase until 2018. The interannual variability is greater for precipitation than for the other variables, but shows no trend on average. Incident radiation also shows a remarkable change around 1988 with about +15 W m$^{-2}$ for direct solar radiation and +5 W m$^{-2}$ for infrared radiation between the periods before and after 1988. At the same time, diffuse solar radiation decreases by 10 W m$^{-2}$ from 1988 onwards. On average, the total amount of solar and infrared energy received by the surface increases by about 10 W m$^{-2}$. This behaviour is consistent with the discussion of Brulebois et al (2015) and the analysis of Boé (2016) and may be caused by several reasons. It can be argued that a decrease in aerosols and the increase in greenhouse gases in the atmosphere have significantly increased incident radiation as shown by climate studies (Wild 2012). In addition to this physical reason, more technical reasons such as changes over time in the density of assimilated observations or changes in the ECMWF operational system may have affected the ERA-40 reanalysis. Although the model used in the reanalysis is a frozen version, the reanalysis system includes input observations whose density varies significantly over time (Uppala et al., 2005). In addition, during the production of the ERA-40 reanalysis, the ECMWF operational data assimilation system has evolved considerably and switched to a 4D-var variational method (1997) compared to the 3D-var method previously used. As a consequence, the calculation of the error covariances of the observations and the guess were revised in the 4D-var but also 3D-var and impacted directly the ERA-40 reanalysis. The comparison in terms of bias and root mean square error (RMSE) at the four weather stations measuring infrared radiation is summarized in Table 2. With the exception of the Carpentras station, where the LSAF IR radiation is almost unbiased and the error is the smallest compared to SAFRAN, the scores are better for the high-altitude stations with SAFRAN when the correction is applied. Due to their high altitude, no correction was applied at Argentière or Saint-Sorlin. At the Argentière station, the bias and root mean square error are lower with SAFRAN than with LSAF. At Saint-Sorlin, the bias is higher with SAFRAN but the RMSE is of the same order of magnitude as LSAF.

**4.2 Impact of new model configurations**

The first comparison concerns the SIM_REF2 experiment where river flow is slightly underestimated compared to SIM_REF (not shown), and the underestimation is corrected by calibrating the subgrid drainage term. In the SIM_REF simulation, the ratio of simulated to observed flow is centered around 1 and the daily efficiency range (NSE below) characterized by its CCDF is larger for all stations. SIM_PHY does not consider any parameterization of the subgrid drainage and is therefore closer to the SIM_REF2 simulation in terms of subgrid hydrology. Figure 5 shows the comparison of each SIM simulation with the observed river flow for the 470 gauging stations. SIM_PHY tends to overestimate flows, as indicated by the average ratio between simulated and observed flows. SIM_PHY shows slightly poorer results for NSE, ranging from 0.5 to 0 (about 40% of the stations), but in this case both models do not perform very well. Most of the stations affected by deterioration in the lower part of the NSE CCDF have an NSE below 0.55 and represent about 57% of the total number of stations. Part of the explanation comes from the calibration of the subgrid drainage in SIM_REF which is not done in SIM_PHY. However, NSE's CCDF shows that SIM_PHY outperforms SIM_REF (and also SIM_REF2, but not shown) for NSEs greater than 0.56, which corresponds to half the total number of stations, and highlights the added value of physics associated with a better description of vegetation types and the use of other, more accurate databases. Figure 5 shows how the scores are improved for experiments with corrected infrared radiation (SIM_FRC), subgrid orography (SIM_TOP) and hydrology (SIM_NEW), both in terms of NSE's CCDF and flow ratio. The bias in river flow is significantly reduced when infrared radiation is increased due to higher total evaporation, resulting in less water available in rivers. However, a positive bias remains, which is expected, since SIM simulates natural runoff and river flow, i.e. without abstraction or diversion, while some basins are influenced by human activity. In some basins, the human footprint on the landscape is characterized by an increase in urban and agricultural areas and the presence of dams. In the model, urban areas have been replaced by rocks, a type of natural surface, to represent the presence of urban areas that enhance surface runoff. However, the model does not explicitly represent irrigation or the impact of the presence of dams on river flow. The basins impacted by human activity are of great interest for the evaluation as they allow quantifying errors in the system and proposing improvements. The SIM_FRC and SIM_TOP NSE scores are very close and better than SIM_PHY for all stations and SIM_REF for about 75% of the stations with NSE greater than 0.4. Finally, SIM_NEW and SIM_TOP tend to overestimate river flow, but their NSEs are significantly better than SIM_REF for all NSE ranges.

Figure 6 presents a map of the differences in mean annual NSEs (for stations with positive NSEs) between the different configurations. Over the entire reanalysis period, in Figure 6a, it is first confirmed that SIM_PHY alone does not improve the flow simulations everywhere in France but only for the gauging stations that were already reasonably represented (with NSEs above 0.56). Second, the new IR forcing improves the scores almost everywhere except in two isolated stations in the Seine basin (Fig. 6b). As expected, SIM_TOP only has an impact on mountains and especially over the Alps (Fig. 6c). Finally, the comparison between SIM_NEW and SIM_TOP highlights the advantages of using an underground mountain

reservoir for snow (Fig. 6d). It should be noted that the number of stations is reduced in Fig. 6c and Fig. 6d because these experiments do not encompass the entire territory. In Fig. 7, SIM_NEW is compared with SIM_REF so that it reveals the advantages of all the changes. The SIM_NEW NSE map indicates that the model explains a large part of the flow variance at most stations (brown to green colours), but some stations still have average (red) to low (blue) NSE values. In particular, the gauging stations in the northern of France are not well simulated, in addition to the Alpine region which is known to have significant anthropogenic influences on the flow regime.

### 4.3 Seasonal river flows

To complement the previous results and to demonstrate the successive improvements in simulated flows, seasonal scores were displayed over the 60-year simulation period using Taylor plots, which have been recognized to be a useful tool for graphically summarizing how a set of simulations compares to observations (Taylor, 2001). A set of experiments can be analysed in terms of correlation, centred root mean square (RMSD) difference and the magnitude of their variation represented by the normalized standard deviation. These scores are calculated from all daily observations and simulations. Seasonal Taylor plots (DJF, MAM, JJA, SON for winter, spring, summer and fall respectively) of the different experiments are presented in Fig. 8. As a result, regardless of the season, the SIM_NEW simulation has the highest correlation and the lowest RMSD, except perhaps for JJA, the season with the highest normalized standard deviation. For DJF, the scores are very good with relatively little spread, while for JJA, the scores are still tightly clustered but the RMSD is higher. MAM and SON confirm the interest of using an underground reservoir to conserve water in the mountains before releasing it in the spring.

### 4.4 Extremes river flows

The previous results showed how SIM_NEW behaved, on average, over the 60-year simulation period. In order to assess the ability of the new system to correctly simulate extreme river flows and thus to distinguish between high and low flow periods, the deciles of daily river flows were calculated and special attention was paid to decile Q10 corresponding to low flow states and decile Q90, the threshold above which a flow is considered to be decadal (here defined as a flood). As shown in Fig. 9, Q10 and Q90 first indicate that in very dry periods (flows less than or equal to Q10), all the simulations except SIM_NEW underestimates the amplitude of the variations. Furthermore, for the SIM_NEW experiment, the correlation, the RMSD and the normalized standard deviation are the best. The variability in terms of normalized standard deviation is reversed when considering floods (Q90) versus dry periods (Q10). Again, SIM_NEW has the smallest RMSD value and all simulation correlations are greater than 0.99. Figure 10 compares the observed and simulated monthly flows of the Garonne River at Lamagistère with SIM_NEW and confirms the model's ability to simulate low flows during the summer seasons fairly accurately and its tendency to overestimate flood peaks.

## 4.5 Snow height

To complement the previous results with respect to flows, a comparison of the snow depths between SIM_REF and SIM_NEW was carried out using the 185 stations described in section 3.2. In Fig. 11, the spatial variability of scores is presented as a function of elevation with notched boxplots where the boxes represent the interquartile range, the whiskers the 10th and 90th percentiles, and the notch represents the 90% confidence interval of the median estimated by a bootstrap sampling technique among the available stations. The SIM_REF simulation has a positive median bias at the lowest elevations and a negative median bias between 2000 and 2400 m, while the SIM_NEW simulation is unbiased at any elevation. The variability of the bias between stations is also reduced in the SIM_NEW simulation. Consistently, a significant reduction in MSE is obtained at the lowest and highest altitudes with SIM_NEW, as well as a reduction in the 90th percentile MSE at all altitudes. These results are consistent with improved altitudinal discretization in mountainous areas, which reduces the altitude differences between the simulated grid cells and the observation stations. Slight improvements in SIM_NEW scores can also be obtained by interpolating linearly the simulated snow depths at the two layers surrounding the observation. However, the point closest vertically to the observation was chosen, in order to use the same selection as in SIM_REF. It should also be noted that improvements in the snow parameterization should also explain some of the improvement in scores (Decharme et al., 2016), but also the use of more accurate vegetation maps.

## 4.6 Changes in the simulated water and energy budgets

This section compares the climatology of the SIM system before and after the changes made. The aim is to qualitatively identify the impact of the new model on the distribution of energy fluxes, which is important for certain hydrological or agriculture-related applications. Maps of the Bowen ratio and the evaporation to precipitation ratio are shown in Fig. 12. The areas with the highest Bowen ratio are located in the mountains where snowfall limits evaporation, along the Mediterranean coast where annual precipitation is lower in quantity and incident radiation rather strong, and in a large area covering the Garonne basin and part of the Loire and Seine basins, characterised by high vegetation fractions. The evaporation to precipitation ratio is also highest in the lowland areas where the Bowen ratio is high. On the mountains, heavy precipitation and limited evaporation due to snow lead to the lowest evaporation to precipitation ratio. These results are comparable to those obtained by Habets et al. (2008) for another period, except that in SIM_NEW, the Landes forest (southwestern France, on the Atlantic coast) has a higher Bowen ratio. The first reason comes from the difference in the parameterization of photosynthesis and more precisely the parameterization of leaf conductance used in SIM_REF based on Jarvis (1976) and SIM_NEW based on ISBA-A-gs (Calvet et al., 1998) which explicitly models photosynthesis (thus the canopy resistance is more physically based) and it models plant stress in a more detailed manner which considerably reduces evaporation over vegetated areas. Thus, the surface energy budget tends to increase the sensible heat flux. The second reason is related to the increase in incoming infrared radiation, which increases the sensible heat flux and decreases the latent heat flux, which generally occurs on dry soils with low evaporation capacity. The interannual variability of the evaporation to precipitation

(E/P hereafter) ratio and the Bowen ratio are presented in Fig. 13, for SIM_REF, SIM_PHY and SIM_NEW to first characterize the old system relative to the new one and to highlight the impact of changes from SIM_PHY to SIM_NEW on the energy budget. E/P is greater in SIM_REF than in the other two simulations each year, and E/P in SIM_NEW is closer to

SIM_REF than in SIM_PHY. Total precipitation is very similar but slightly lower in SIM_REF and in SIM_PHY or SIM_NEW, due to the representation of sub-grid orography in the mountains, enhanced by a higher resolution of the orography which allows for finer vertical discretization. Therefore, higher E/P corresponds to higher total evaporation. In SIM_NEW, the ratio of simulated to observed flow is in excess whereas it is better simulated in SIM_REF with a peak centred around 1. This result is consistent with an evaporation deficit in SIM_NEW compared to SIM_REF. The Bowen

ratio is lowest for SIM_REF, increases in SIM_PHY, and is highest in SIM_NEW, which already tends to evaporate more than SIM_PHY. This result shows that the sensible heat flux in SIM_NEW is much higher than in SIM_PHY, mainly due to the increased incoming infrared radiation, which partially compensates for the evaporation deficit.

## 5. Discussion

### 5.1 Climatic data

As shown in Fig. 4, there is heterogeneity in the forcing data, particularly with respect to radiation. There are two possible reasons for the break in the time series, the first is due to the large-scale analysis used to reconstruct temperature, humidity and cloudiness profiles. As explained in Section 4.1, the calculations of these profiles have varied over time as a result of improvements in the global data assimilation systems used in the ERA-40 reanalysis production. The second reason is the variation in the observation density network over time. Indeed, from 1958 to the present, substantial changes have been

observed in the deployment of new weather stations. The combination of these two changes means that the SAFRAN reanalysis is not homogeneous over time and it seems important to understand how the Optimal Interpolation results are influenced by these changes when analysing the simulation results. However, an abrupt change may also be due to the darkening / lightening effect (Wild 2012, Brulebois et al., 2015, Boé 2016).

As already mentioned, the uncertainty in SAFRAN's IR radiation is significant. The ability to observe the IR in the plains

and mountains allowed a fair comparison between LSAF and SAFRAN products without correction (SIM_PHY) and with correction (SIM_FRC). The impact of this variable is very important, especially over snow (Quéno et al, 2017, Sauter and Obleitner, 2015), therefore, an extension of the in situ observation network would allow a better understanding of its spatial variability and the potential improvement of model simulations. The extension of the correction to the entire French territory is debatable, but this decision was guided by the positive bias of river flows and also by the desire to have a more realistic

energy input in mid-mountain areas (i.e. below 1500 m) in order to better model the evolution of the snowpack.

We also compared the simulated soil temperatures to the observations made over France. The IR correction on soil temperature has a positive impact and significantly reduces biases and RMSEs (not shown). The results are consistent and of the same order of magnitude as those obtained by Decharme et al (2013).

### 5.2 River flows

The results show that SIM_REF simulates the correct ratio between modelled and observed river flow (centred around 1) whereas in SIM_PHY, this ratio indicates an overestimation. However, errors in the forcing data show that errors compensate for each other in SIM_REF, since despite a radiative deficit, river flow is rather well simulated. In SIM_PHY, as explained in the model description, more complexity has been added to the model based on a better representation of physics. The calculations, performed on each of the vegetation types, use the A-gs photosynthesis parameterization, which

tends to produce less evaporation on the vegetation, leading to more water available in the rivers. On the other hand, it has already been mentioned that radiative forcing is underestimated. The combination of more water available in the soil and less radiative energy to evaporate leads to an overestimation of river flows. By correcting for IR radiation, the SIM_FRC simulation shows a clear improvement in river flow scores, with a peak of the modelled to observed ratio closer to 1, and an improved daily efficiency range in almost all cases, except perhaps for NSEs below 0.4, but in this case the difference with

SIM_REF is very small. The implementation of the subgrid topography with the use of elevation bands (SIM_TOP) and the subgrid hydrology with the inclusion of a snow reservoir (SIM_NEW) essentially impacts the hydrology in the mountains, and thus the snow and river flows that are affected by snowmelt.

### 5.3 Snow depth

For the evaluation of the snow depth, the comparison can only be made on the 9892 cells, which corresponds to the

SIM_REF grid. In addition, in order not to disadvantage SIM_REF and to assess the impact of changes in physics and atmospheric fields, sub-grid processes in SIM_TOP and SIM_NEW were not considered in the evaluation (the additional vertical levels of the 1044 cells were not used). Thus, the snow depth simulated in SIM_FRC, SIM_TOP and SIM_NEW is the same because all these experiments use the same correction for infrared radiation. In terms of snow depth, only SIM_PHY would be different from the other three simulations if the IR correction were to be applied below 1340 m, which

limits the interest of such a comparison.

### 5.4 Subgrid hydrology

This method showed that the hydrology of mountainous areas was improved because the analysed precipitation rate and phase were better represented for each altitude band than when averaged vertically, resulting, in the case of the Durance

River (Lafaysse et al., 2011), in a decrease in the overestimated spring peak flow associated with a better phase between the observed monthly flow and the simulated flow. However, summer and winter peak flows were still significantly underestimated by the model. During long periods of drought without precipitation or snowmelt, river flows are controlled by subsurface drainage. In the framework of the Aqui-FR project (https://www.metis.upmc.fr/~aqui-fr/) aimed at developing a platform with multiple regionally-specialized hydrogeological models over France to simulate flows and water table

heights, aquifers are explicitly simulated and the water flows of SURFEX (Masson et al., 2013) used as inputs should not be impacted by an empirical representation of aquifers. Moreover, in Aqui-FR, some hydrogeological applications have been calibrated using SURFEX runoff and infiltration water flows as inputs (Vergnes et al., 2019).

## 6. Conclusions and outlook

This study illustrates how developments over the last ten years are improving the SIM hydrometeorological system. Several
important changes have been made, particularly in the soil physics of the ISBA model where the force-restore method has been abandoned and replaced by the multi-layer soil diffusion method. At the same time, as described in section 2, the snow model has been revised to improve vertical layering, snow compaction and solar energy transmission within the snowpack through the use of spectral albedo, as it is done in more advanced models.  The model was run according to the vegetation tiling approach, with each of the 12 vegetation types characterized by its own set of parameters, in contrast to the single
vegetation type approach where the parameters are aggregated. Then, more accurate databases for soil, orography and land use were used. A more precise infrared forcing significantly improved the results as well as the use of a groundwater reservoir in mountains associated with a specific vertical discretization of the massifs. The new configuration of the model, including all the new or updated functionalities mentioned above, proved to be more efficient than the old system and was therefore better adapted to water resource studies. Comparisons with independent observations of daily total snow depth and
river flows were made and confirmed that the scores were improved. In addition, the new SIM system better represented river flow extremes for both low and high flow periods.

Some perspectives can be proposed to improve the SIM system. The first is to improve the description of climate. It was found that SAFRAN worked well in most cases, but some shortcomings remained. A new near-surface reanalysis system is being developed at Météo-France to replace SAFRAN. It includes a new surface analysis of air temperature, relative
humidity at 2 meters and daily precipitation, and uses high-resolution model outputs as first guess of the analysis. In addition, as part of the Copernicus program, a 5.5 km high-resolution reanalysis will be produced over Europe, and will be an interesting product to compare with SAFRAN over France.

The second is to improve the representation of surfaces in the model. Indeed, the ecosystem database is representative of the 1999-2006 period. For more recent simulations, or quasi-real-time applications, it would be interesting to study the
contribution of new high-resolution satellite products, such as the land cover product of the European Space Agency and the Climate Change Initiative, or certain other parameters derived from Copernicus products, such as albedo for example, which allow a better description of surface types.

The third concerns improving the physics of the model and more specifically the use of the multi-energy balance (MEB) scheme (Boone et al., 2017; Napoly et al., 2017) to enable explicit calculation of the interactions of the canopy with the air
and the ground. The MEB model showed some modest gains within the SIM_REF simulation, owing to a better temporal

partitioning between bare soil evaporation and transpiration (Napoly, 2016). Moreover, the MEB model demonstrated that the use of litter in forests improved surface flux results.

Secondly, considering anthropisation, in particular irrigation and the presence of dams, could benefit the SIM system to improve its realism and allow more accurate comparisons with gauging stations in anthropized basins. Irrigation is currently

developed in the ISBA model and the integration of dams is a longer-term project. Finally, a better representation of groundwater and its characteristics in France is another challenge to be taken up.

*Code availability*. The SURFEX v8.0 source code, including the ISBA code, used in this study is available in the supplement, as well as the SAFRAN code. The post-processing codes, including the *scores* package from the open source

*snowtools* project, are also available in the supplement.

*Data availability*. The results of all the models examined here and the R and Python programs for plotting the results are available in the supplement.

*Supplement*. The supplement for this article is available online at http://doi.org/10.5281/zenodo.3689560.

*Author's contribution*. Model developments were performed by PLM and EM for the SAFRAN analysis on France, SF for SURFEX and BD and AB for the diffusive version of SURFEX/ISBA, and FH for MODCOU. PLM and FB designed the experiments and carried them out. ML carried out the comparison of the results on snow. DL provided valuable Python scripts for the figures. JB and ML first tested the model for their own research. PE, FB and FR are responsible for the SIM operational suite at Météo-France. PLM prepared the manuscript with the help of all co-authors.

**Appendix A**

The formula of the infrared correction

This correction was proposed to compensate for a deficit in long-wave radiation analysed by SAFRAN compared with infrared measurements from two reference meteorological stations, Carpentras and Col de Porte, located respectively in south-eastern France and the Alps. The correction is applied below 1340 m. The comparison was made for measurements

collected between August 1993 and August 1994 every three hours.

The correction is written as follows:

$$\varepsilon(\sigma) = (-5.42 + 1.14\sigma - 0.11\sigma^2) * 10^{-2} \quad (A1)$$

$$LWD_{cor} = LWD_{ref}/[1 + \varepsilon(\sigma)] \quad (A2)$$

Where $\sigma$ is the cloudiness analysed in octas, $LWD_{ref}$ the SAFRAN longwave downward radiation, and $LWD_{cor}$ the

longwave downward radiation when the correction is applied, i.e. when it is divided by $1 + \varepsilon(\sigma)$. Figure A1 shows the

magnitude of the correction as a function of cloudiness. The increase in radiation is highest under clear sky conditions and decreases with cloudiness up to 5 octas and increases again for cloudier skies.

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

**Table 1: main characteristics and differences in experiments.**

|  | *SIM_REF* | *SIM_REF2* | *SIM_PHY* | *SIM_FRC* | *SIM_TOP* | *SIM_NEW* |
|---|---|---|---|---|---|---|
| ***Land Surface Model*** |  |  |  |  |  |  |
| Soil transfers | Force-restore | Force-restore | Diffusion |  |  |  |
| Soil layers | 2 or 3 | 2 or 3 | 14 soil layers |  |  |  |
| Snow layers | 3 | 3 | 12 snow layers |  |  |  |
| Photosynthesis | No | No | A-gs module |  |  |  |
| Vegetation types | 1 | 1 | 12 |  |  |  |
|  | b=0.5 | b=0.5 | b=0.25 |  |  |  |
|  | Calibrated | Forced to 0. | No |  |  |  |
| ***Hydrology*** |  |  |  |  |  |  |
| Subgrid runoff | ECOCLIMAP | ECOCLIMAP | ECOCLIMAP2 |  |  |  |
| Subgrid drainage | 1 | 1 | HWSD |  |  |  |

| Databases | INRA | INRA | SRTM90 | | | |
|---|---|---|---|---|---|---|
| Vegetation<br>Soil<br>Topography | GTOPO30 | GTOPO30 | | | | |
| **Infrared radiation**<br>Correction | Off | Off | Off | On | On | On |
| **Mountain specificity**<br>Subgrid topography | Off | Off | Off | Off | On | On |
| Drainage reservoir | Off | Off | Off | Off | Off | On |


**Table 2: Annual mean BIAS and RMSE of LSAF, SAFRAN and corrected SAFRAN infrared radiations at Carpentras (95 m), Col de Porte (1340 m), Argentière (1900 m), and Saint-Sorlin (2620 m).**

| | LSAF IR radiation $W\,m^{-2}$ | | SAFRAN IR radiation without correction $W\,m^{-2}$ | | SAFRAN IR radiation with correction $W\,m^{-2}$ | |
|---|---|---|---|---|---|---|
| | *BIAS* | *RMSE* | *BIAS* | *RMSE* | *BIAS* | *RMSE* |
| **Carpentras** | 1.3 | 10.2 | -8.4 | 21.5 | 3.1 | 20.3 |
| **Col de Porte** | -10.3 | 20.2 | -14.4 | 20.4 | -9.3 | 17.5 |
| **Argentière** | -18.8 | 32.6 | -3.6 | 18.5 | -3.6 | 18.5 |
| **Saint-Sorlin** | 0.1 | 27.8 | 10.5 | 25.3 | 10.5 | 25.3 |

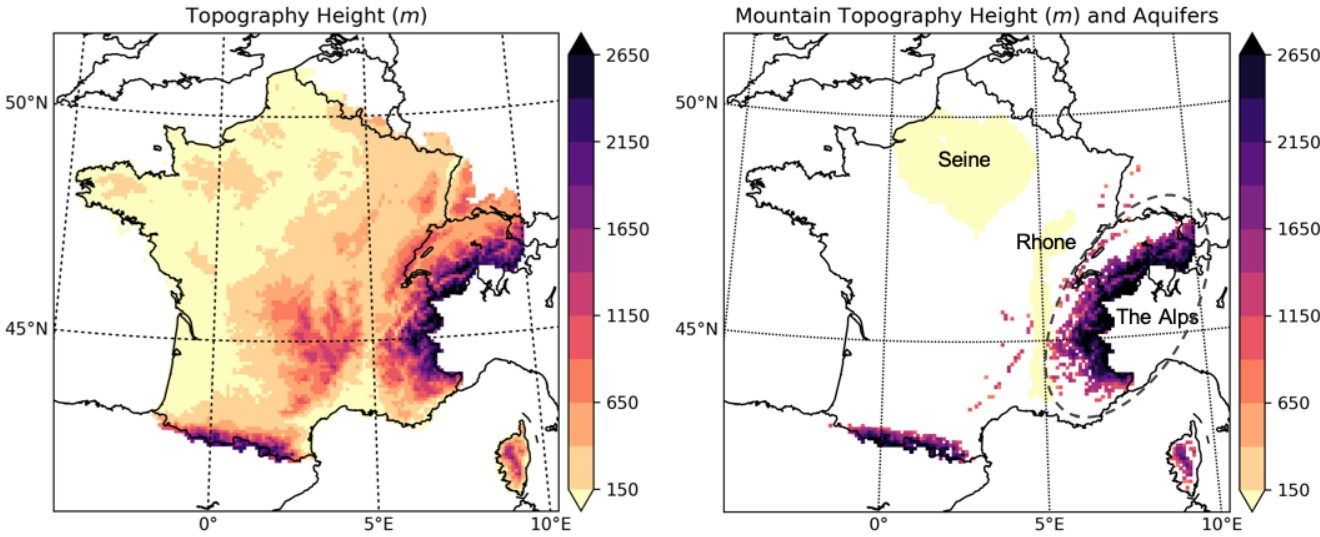


**Figure 1: Height of the topography of the 9892 cells of the SIM grid (left) and the 3878 cells of the mountain SIM grid (right). The cells of the mountain grid correspond to the 1044 points having an altitude greater than 500 m and described vertically by several layers. Zones in yellow correspond to the Seine and Rhone aquifers. The dotted line delimits the Alps mountain.**


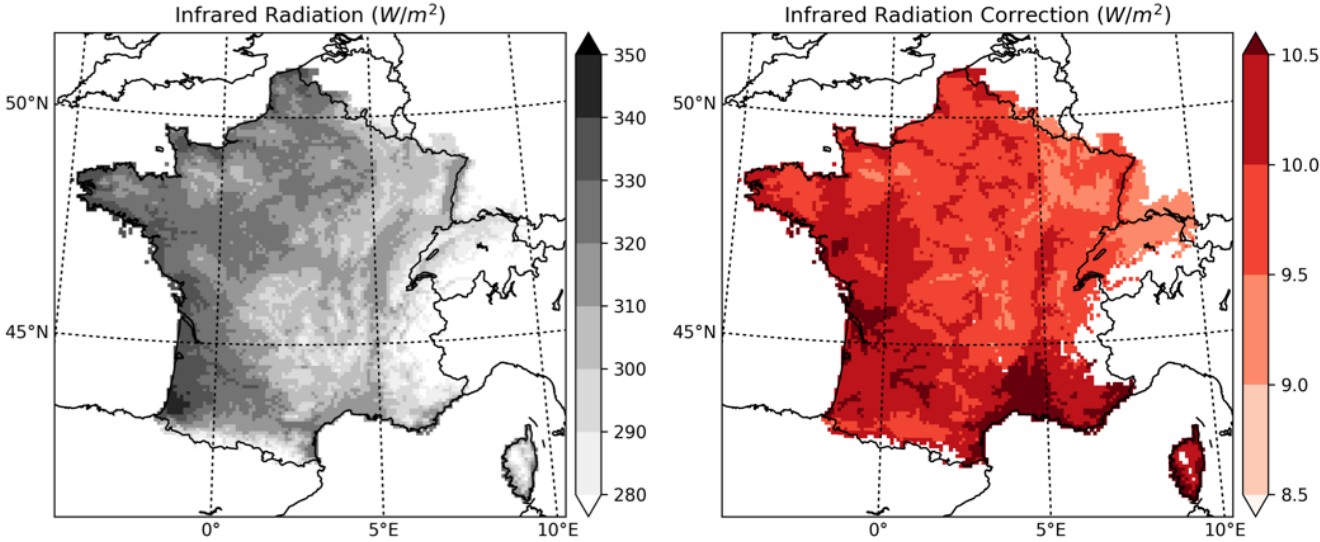

**Figure 2: Annual average of uncorrected (left) and corrected (right) downward longwave infrared radiation from SAFRAN analysis.**

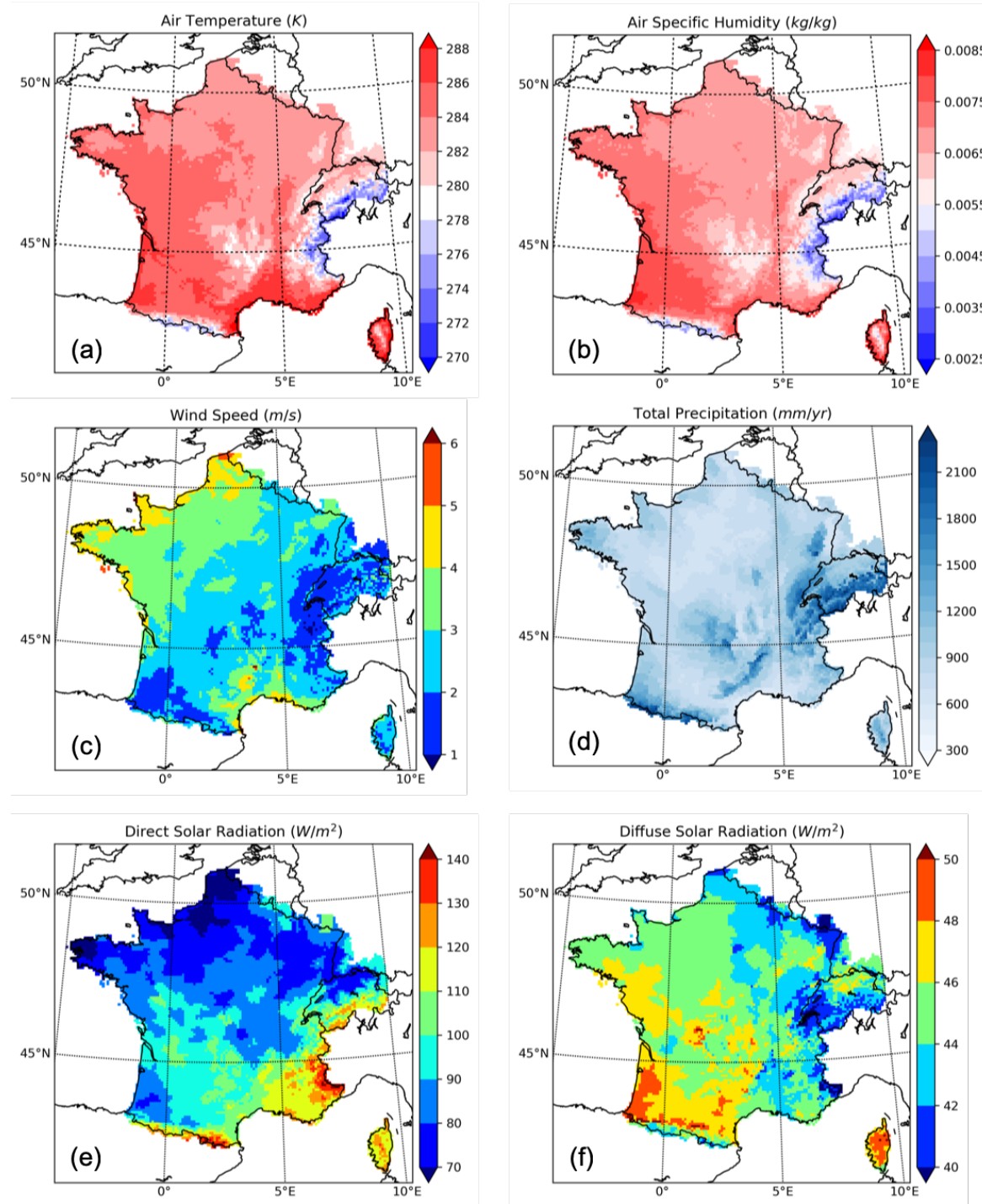

**Figure 3: Maps of annual average of the SAFRAN analysis for the period 1958-2018 of (a) air temperature at 2 meters, (b) specific air humidity at 2 meters, (c) wind speed at 10 meters, (d) total annual precipitation, (e) direct solar radiation, and (f) diffuse solar radiation.**

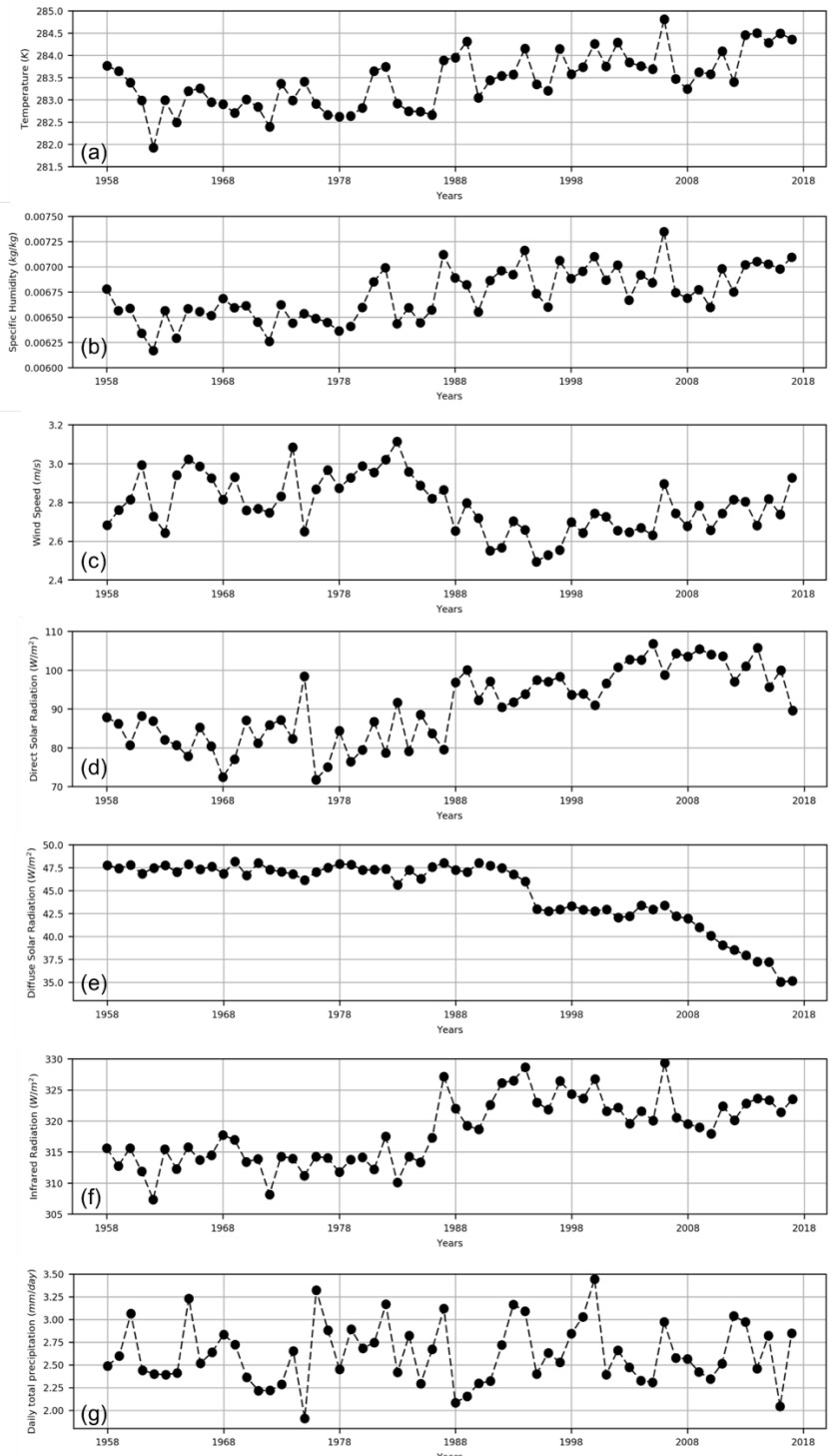

**Figure 4: Annual average of the SAFRAN analysis of (a) air temperature at 2 meters, (b) specific air humidity at 2 meters, (c) wind speed at 10 meters, (d) direct solar radiation, (e) diffuse solar radiation, (f) infrared radiation, and (e) total precipitation rate.**

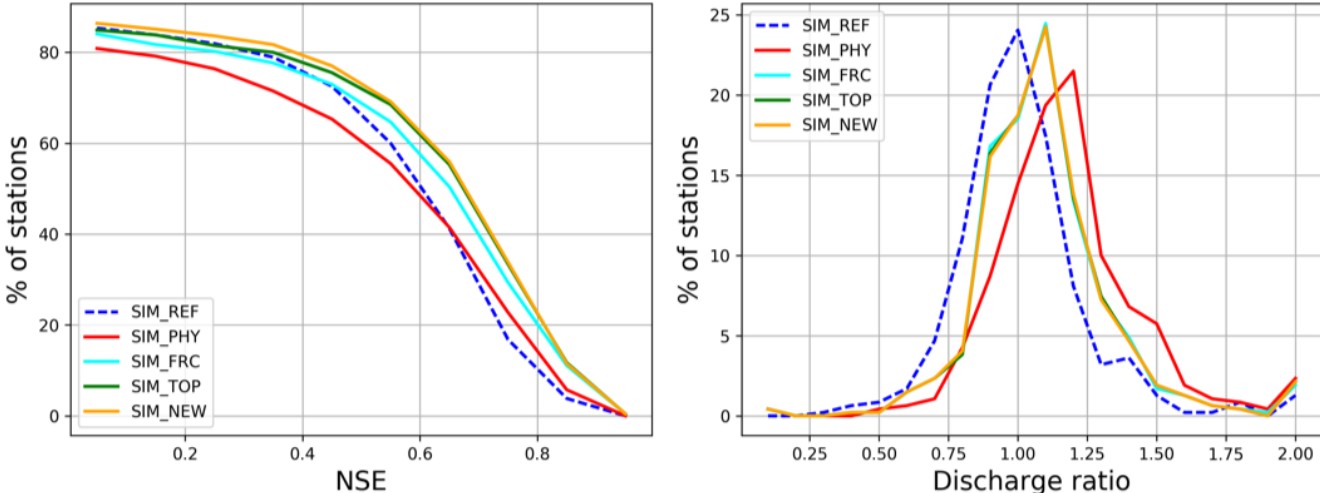

**Figure 5: Comparison of the NSE CCDF (left panel) and the simulated to observed flow ratio (right panel) for SIM_REF (dashed blue line), SIM_PHY (solid red line), SIM_FRC (solid cyan line), SIM_TOP (solid green line), and SIM_NEW (solid orange line).**

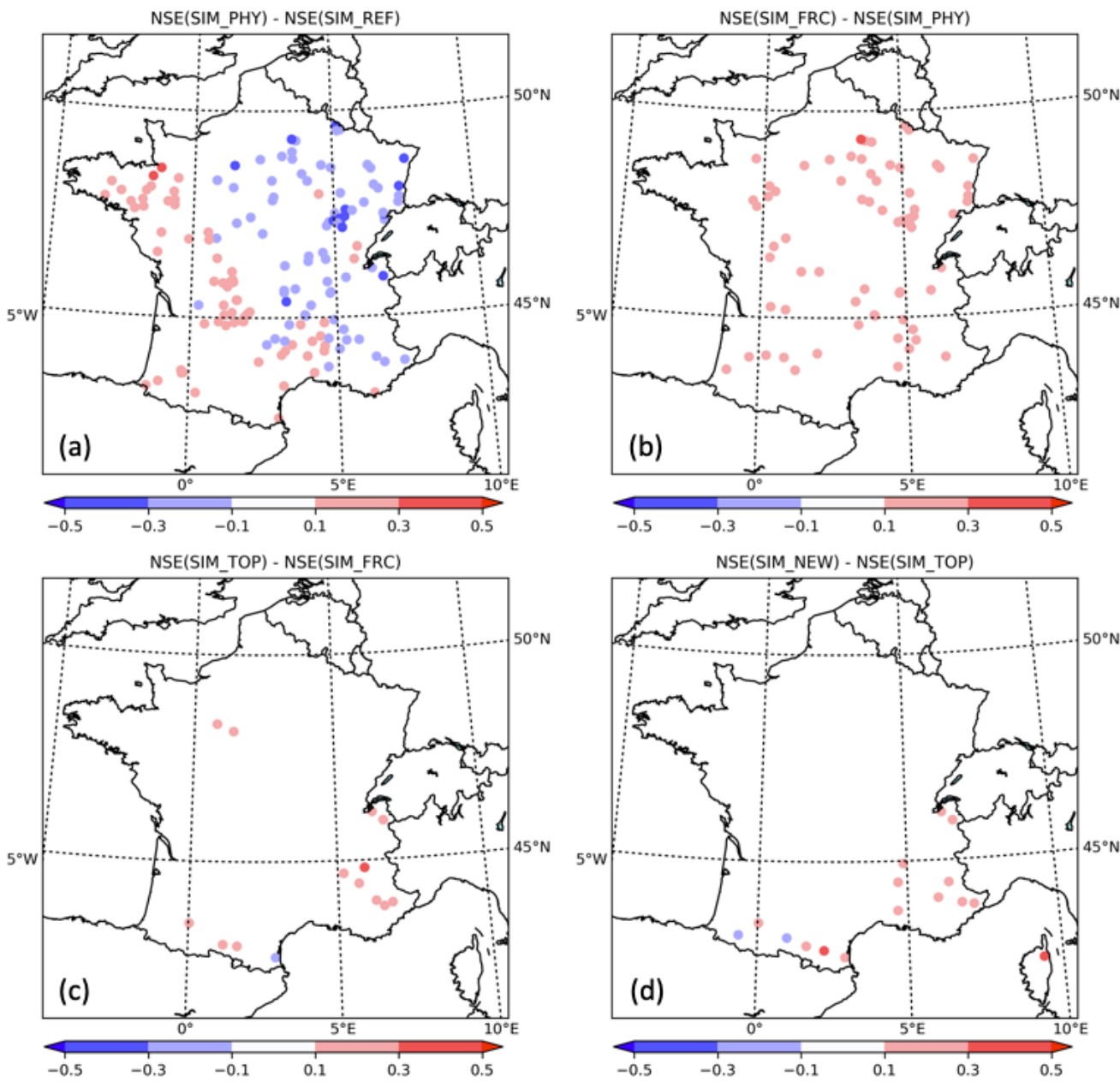

**Figure 6: Maps of the difference in mean NSE for NSE>0 between simulations: (a) SIM_PHY and SIM_REF, (b) SIM_FRC and SIM-PHY, (c) SIM_TOP and SIM_FRC, (d) SIM_NEW and SIM_TOP.**

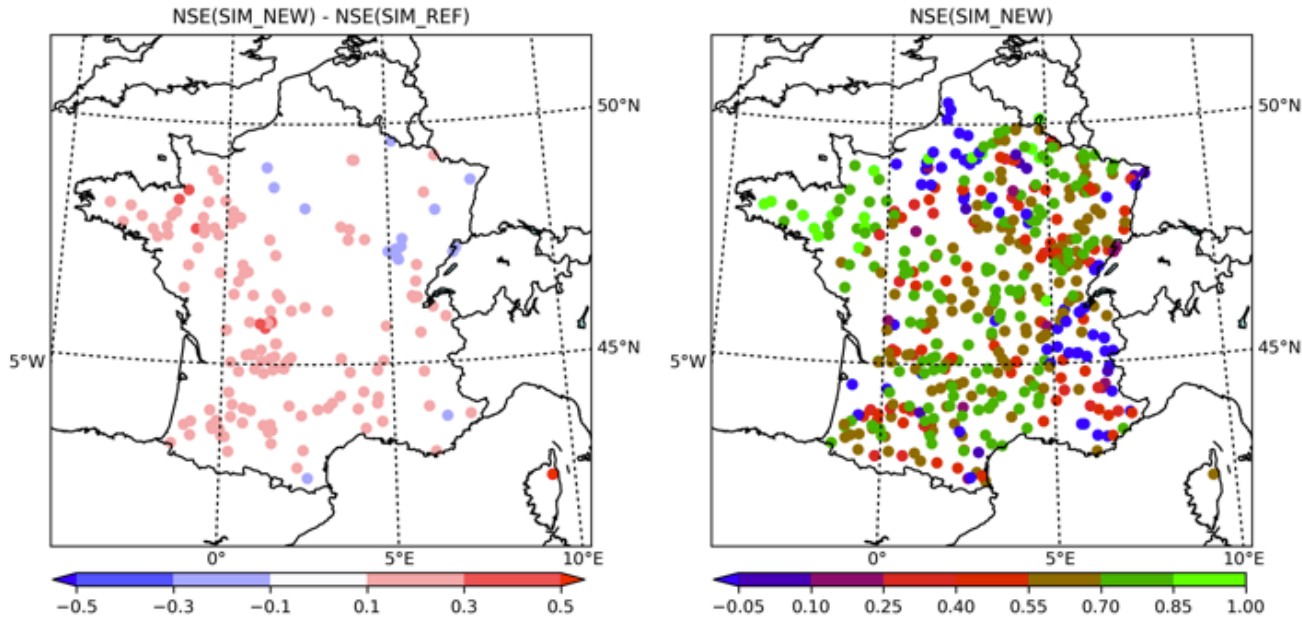

**Figure 7: Map of the difference in mean NSE for NSE>0 between SIM_NEW and SIM_REF (left panel), and SIM_NEW NSE map (right panel).**

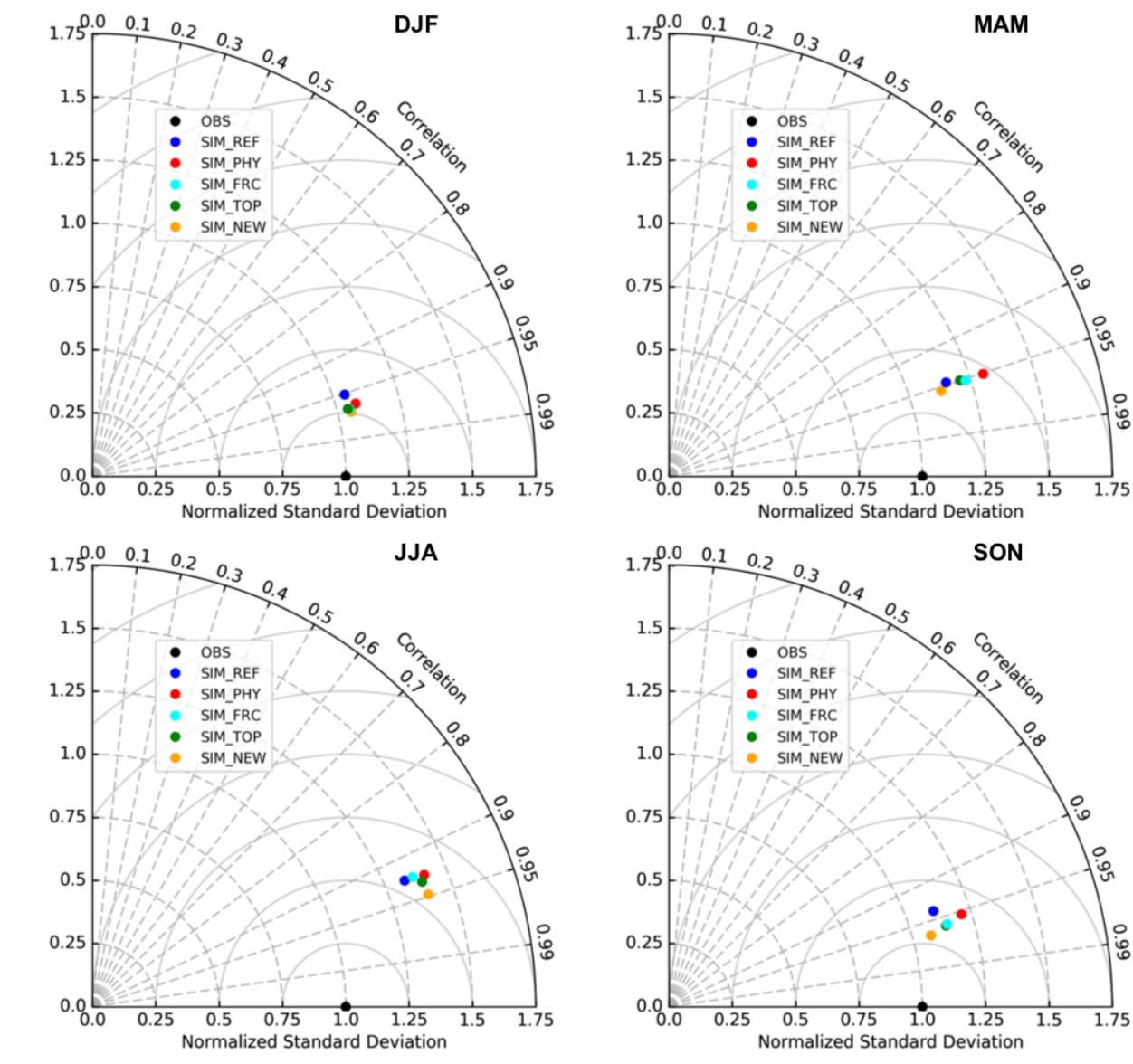


**Figure 8: Taylor diagrams of seasonal river flows for the different experiments over the period 1958-2018.**

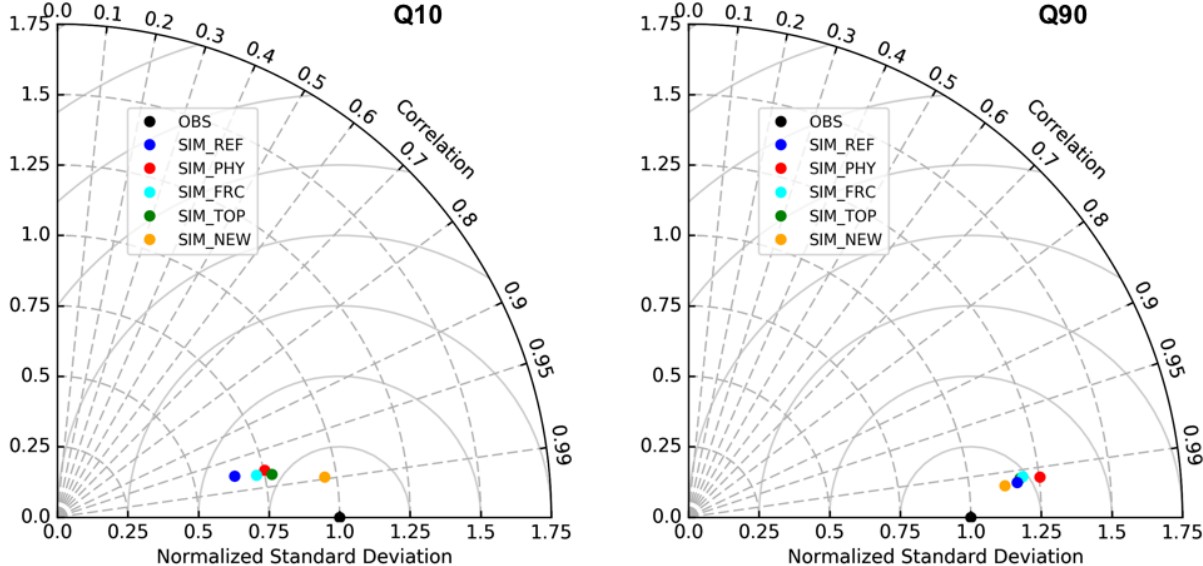

**Figure 9: Taylor diagram of Q10 and Q90 deciles of river flows over the period 1958-2018.**

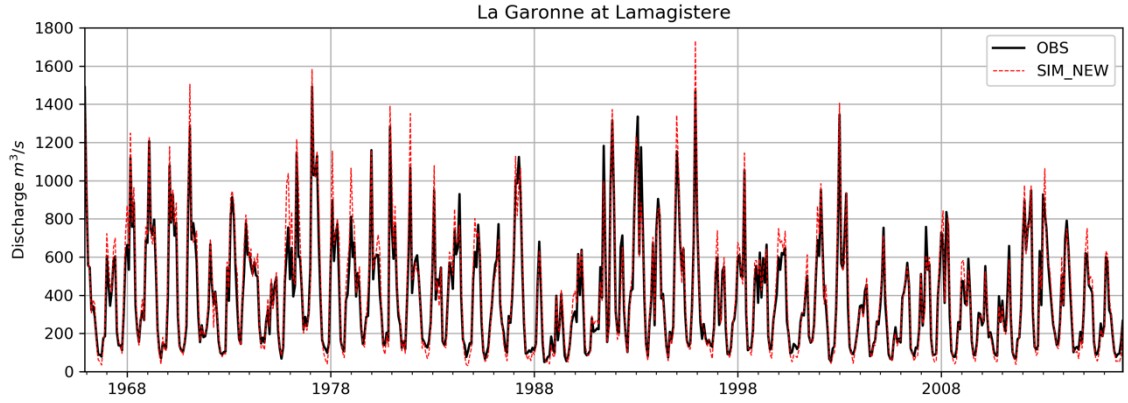

**Figure 10: Comparison of monthly river flows with SIM_NEW for the Garonne at Lamagistère over the period 1958-2018.**

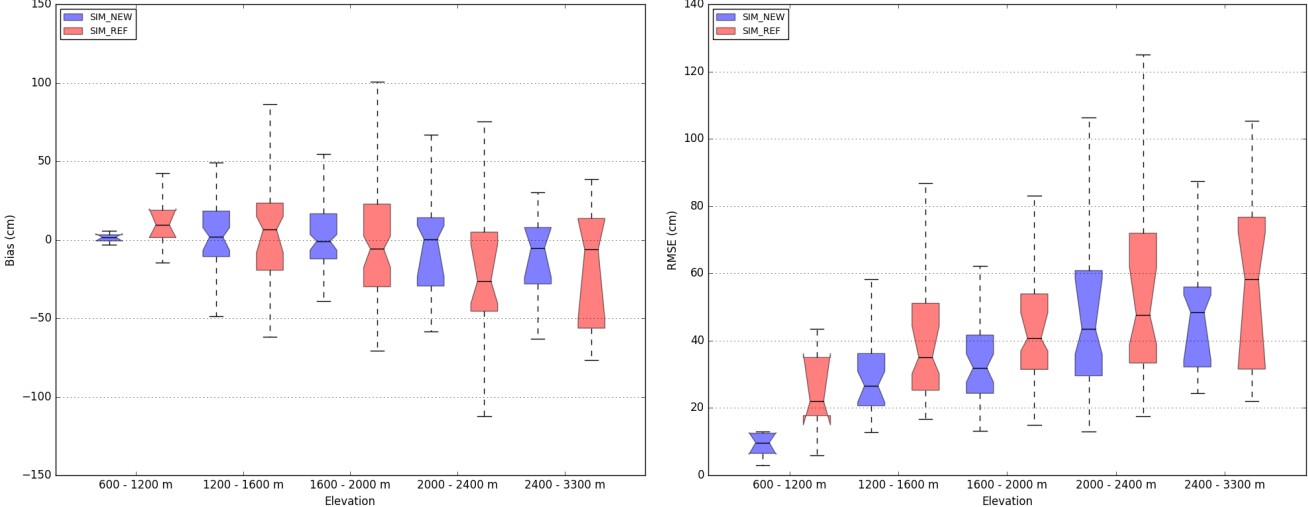

**Figure 11: Bias and RMSE of daily total snow depth for SIM_NEW (blue) and SIM_REF (red) simulations, as a function of elevation. The scores are computed for 185 stations over the period 1984-2016 for months between October and June. The boxes represent the interquartile interval, the whiskers the 10th and 90th percentiles, and the notch represents the 90% confidence interval of the median estimated by a bootstrap sampling technique among the available stations.**


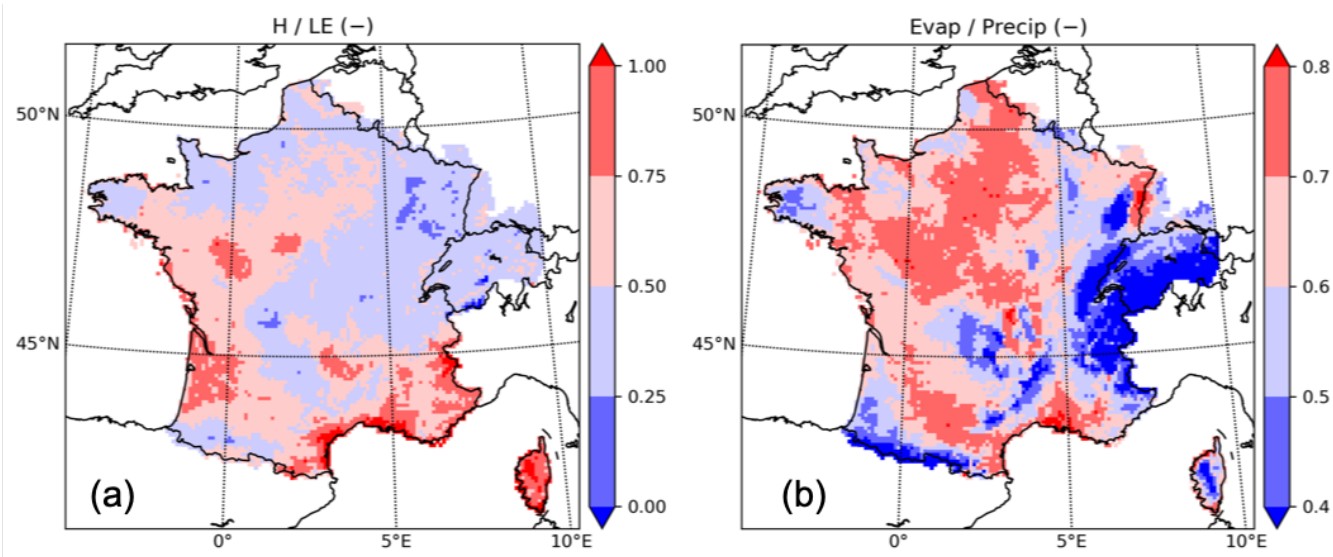

**Figure 12: Maps of mean annual Bowen ratio (a) and evaporation to precipitation ratio (b) for SIM_NEW on average over period 1958-2018.**

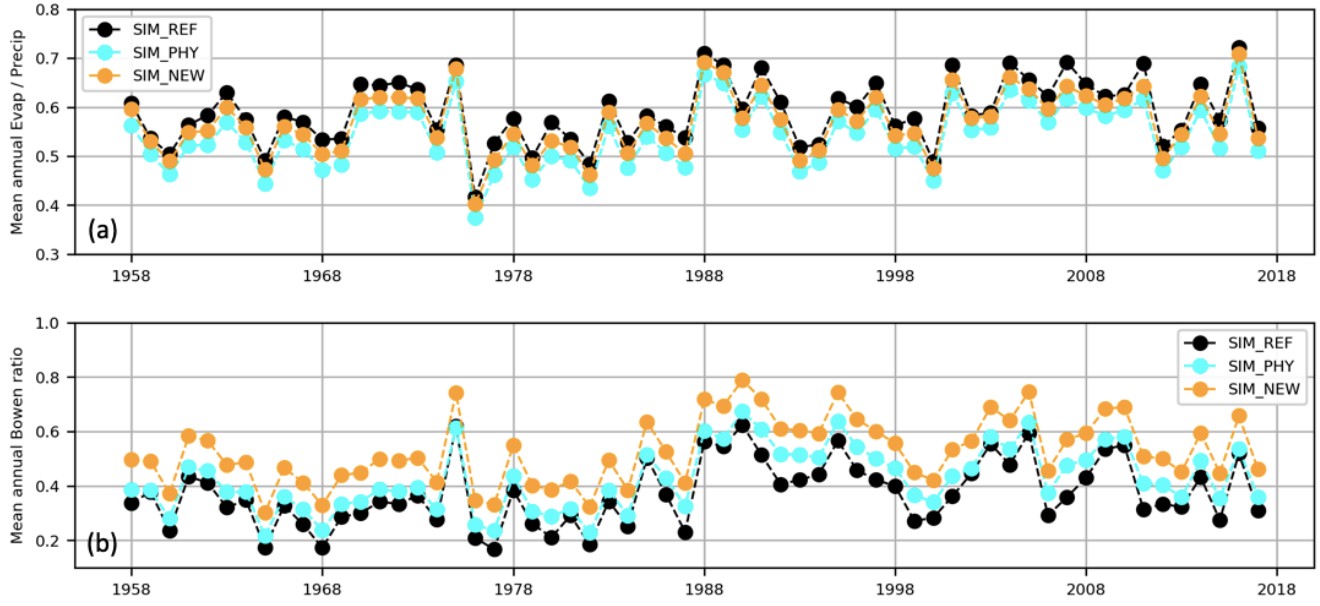

**Figure 13: Mean annual evaporation to precipitation ratio (a), and Bowen ratio (b), for experiments SIM_REF, SIM_PHY, and SIM_NEW.**

