# Peer review of "The latest improvements with SURFEX v8.0 of the Safran-Isba-Modcou hydrometeorological model for France"

_Geoscientific Model Development, 2020_

## Referee Comment (RC1) · Anonymous Referee #1 · 10 Mar 2020

This paper presents an analysis of a number of improvements to a land surface model. I do appreciate the amount of work the authors have performed, but at this point I also have a number of major comments to the paper.

- A first problem is the quality of the writing. There are a large number of grammatical errors that many times make the paper very hard to read. This renders the paper unacceptable for publication.

- Top of page 2: There is a vast amount of literature on the evaluation of land surface models at various spatial scales using Bowen ratio or eddy covariance data (latent and sensible heat fluxes), ground heat flux data, soil temperature and moisture profiles,

infiltration data, ground water levels, etc. Similar comparisons have been performed using remote sensing data. Stating that models have been evaluated using discharge data and ground water levels is blatantly ignoring an enormous amount of literature.

- We need more information on how the point data were upscaled to allow comparisons with the model grid results. Comparing point data to spatially averaged model results does not make much sense. Furthermore, some grids may contain more stations than others. This will impact the results of the comparison, and needs to be explained better.

- Please make it crystal clear how the 470 discharge stations were selected. If there are more stations, why were they not used?

- Line 351: some basins are anthropized. I assume this means "urbanized", because agricultural crops are now in the new vegetation classes. If basins are urbanized or semi-urbanized, and the model does not simulate this, then these basins cannot be used to evaluate the model.

- Line 417: "can probably be attributed to a deficit in the incoming shortwave ration ... or geothermal heating". This seems speculative, it could simply be that the soil or land cover parameters are wrong.

- Section 4.7: remote sensing data could have been used to substantiate the results here. At this point, unless I am misunderstanding, no data are used to validate the conclusions.

In the light of these comments I unfortunately cannot recommend the paper for publication in a highly esteemed journal like GMD.

Some minor issues:

- Add units to the list of numbers at the bottom of page 3.

- Top of page 4: be specific, do not state "few tens of centimeters" and "a few meters".

- Line 215-220: this explanation is very unclear.

- Section 4.7: Bowen ration -> Bowen ratio.

- There are 17 figures in the paper. This seems like a bit much to me. Can this perhaps be reduced?

**[GMDD](https://doi.org/10.5194/gmd-2020-31)**

---

## Referee Comment (RC2) · Anonymous Referee #2 · 10 Apr 2020

Summary:

This paper describes the new developments in the Safran-Isba-Modcou (SIM) hydrometeorological model as part of SURFEX v8.0. These new developments cover different aspects of the SIM system: atmospheric forcing, climate fields, land-surface model parameterizations and water budget parameterizations. The authors evaluate the impact of the new developments in an incremental way using different types of observations. They show that the new SIM system, considering all model changes, improves the simulation of daily river discharges and snow depth over a set of catchments and sites covering the French region, compared to the previous SIM system.

[Figure]

The discussion of the results is clear in most of the parts and conclusions are overall justified by the results shown. However, in my opinion the authors do not fully discuss the interactions among the different model changes, therefore explaining the physical mechanisms of some of the results. Also, I found the section on the soil temperature evaluation weak compared to the others. Finally, I have few comments on the introduction and model description parts: these sections are a bit difficult to read and can be improved. All these points are discussed in the main comments below. In summary, I would recommend the acceptance of the paper after major revisions, to make the paper stronger and more attractive to readers.

Main comments

- Introduction and model description: Sect. 1 and Sect. 2 are quite difficult to read. Sentences are not clear in some places, making it difficult to understand the message that the authors want to deliver. I would suggest the authors to improve the readability of these sections. Few examples are reported in the minor comments below.

- Discussion of the results: The authors performed a set of sensitivity experiments to extract the effect of each model change. However, I have the impression that they do not fully discuss the interactions among the different model changes, and how these relate to the results of the evaluation. For instance, why the SIM_PHY simulation deteriorate the scores in many of the presented metrics (Fig.8, Fig.9, Fig.11)? Is it because of errors in other components, like the atmospheric forcing, which then penalise a more physically complex model? If that is the case, are the other changes reducing such errors, therefore allowing to fully exploit the benefit from the new soil/snow parameterizations? Or are these unrelated? Another example is in the evaluation of the snow depth: the authors propose several hypothesis to explain the improvement of the simulation of the snow depth in SIM_NEW. However, the comparison of all experiments would clearly quantify which processes/changes are responsible for the improvements (see also one of the minor comments below).

- Soil temperature evaluation: I found Sect. 4.6 on the analysis of soil temperature profiles rather weak and with not enough details. The scores of the new system, without a reference, cannot be put into the context of the paper and so are not adding valuable information to the results. A comparison between the different simulations would clarify at least the impact of each change on the bias. The authors state that such biases can be associated to incoming shortwave radiation or lack of geothermal heating, but what about the soil parameterization or soil/surface properties?

Minor comments

Abstract: The main scope of the paper is placed at the very end of the abstract. This could be placed earlier in the text to make clearer the main message of the paper.

Ln-35–40: This paragraph should be rephrased and clarified. Also references to previously published work on the evaluation of land surface and hydrological models should be introduced in the text.

Ln.42: What do you mean by "independent" variable? As the authors stated few lines before, surface energy and water budgets form a coupled system. Please clarify/reformulate.

Ln. 49-50: "... , where modelling contribution of SVAT . . . accounted for in models." this sentence is not clear, please reformulate.

Ln.65-73: The scope of the work can be improved, to make it more precise and easier to read. For instance the authors talk about "new parameterizations" at line 66, but changes to the atmospheric forcing (Ln. 68) are not a parameterization. Also at Ln.69 they talk about "these results", but it is unclear which results are referring to at this point.

Ln. 83: could you be more specific on the horizontal resolution of the SAFRAN analysis?

Ln. 83: is the 24h precipitation analysed every 6-hours?

Ln. 154: please clarify in which sense you mean "dry", as this can be related to various processes.

Ln. 163-167: this paragraph should be simplified/clarified, to make it clearer that the soil map is not changed between the previous SIM system and the new system (as far as I understood).

Ln. 166-167: please clarify that this map was the one used in the "old" version of SIM.

Sect. 2.3: as far as I understood, the main change in the climate fields is the update to ECOCLIMAP2. Hence, I would suggest to expand the discussion of this change, for instance explaining the impacts on the ISBA model? See also next comment.

Ln. 187-193: this sentence is very long. Coud you break it in multiple sentences, better explaining the impact of this change?

Sect. 2.4, title: could you specify in the Section title that this is downward infrared radiation?

Ln. 195-198: is the bias related to a problem in the analysis (for instance cloud cover) or a RT model issue?

Ln. 214: annual mean over which time period?

Ln. 215-217: could you clarify this sentence? is the analysis done every 300m in the vertical direction?

Ln. 233: why between 3 to 5 layers are necessary, and not more? What it is the vertical discretization between each band? Please clarify.

Ln. 264: How does the relatively low horizontal resolution of the ERA-40/ERA-I data impact the simulations? The horizontal resolution of ERA-I is $\sim$80km, that is one order of magnitude less than the one used by the SIM grid. I am thinking for instance at regions with a low coverage of surface stations used in the analysis.

Ln. 299-300: were the data cleaned in some way? For instance removing stations with a few number of observations? Or all data have been used to compute the statistics? The latter could introduce some artifact in the statistics. This should be better explained in the text.

Ln.318-320: could you specify clearly in the text when the transition from ERA-40 to ERA-I occur in SAFRAN?

Ln. 335-345: What is the reason for the deterioration in the lower part of the CCDF of NSE in SIM_PHY?

Sect.4.3: I would suggest to rename this subsection as it is quite vague at the moment: most of the paper regards the comparison to old SIM.

Ln.405: why not adding a third box for SIM_PHY to evaluate the effect of the new snow/soil schemes on the snow depth?

Ln. 413: "baresoil"–> bare soil

Ln.414: At which depth the soil temperature observations are taken? Is any interpolation applied to the data?

Ln.440-449: It would be nice to explicitly link this discussion on the changes of Evap/Precip with the changes in the discharge mean bias.

Ln. 489: I would rephrase this sentence for readers not familiar with detailed snow models.

Comments on the figures

- Generally, the figure captions should be improved to make them more self-explanatory.

- In the caption of Figure 1, the definition of a "mountain grid cell" should be added.

- Some of the figures could be merged together, for instance Fig. 7 with Fig. 8 and Fig.

9 with Fig. 10, for conciseness.

---

## Author Comment (AC1) · 27 Apr 2020

- The colour black is used for the reviewer questions/remarks.
- The colour green is used to answer questions raised by the reviewer.
- The colour blue is used for the text added to the manuscript.

This paper presents an analysis of a number of improvements to a land surface model. I do appreciate the amount of work the authors have performed, but at this point I also have a number of major comments to the paper.

- A first problem is the quality of the writing. There are a large number of grammatical errors that many times make the paper very hard to read. This renders the paper unacceptable for publication.

- The manuscript was completely revised and corrected by a native English speaker.

- Top of page 2: There is a vast amount of literature on the evaluation of land surface models at various spatial scales using Bowen ratio or eddy covariance data (latent and sensible heat fluxes), ground heat flux data, soil temperature and moisture profiles, infiltration data, ground water levels, etc. Similar comparisons have been performed using remote sensing data. Stating that models have been evaluated using discharge data and ground water levels is blatantly ignoring an enormous amount of literature.

- It was not intended to state that the models were evaluated using only flow and groundwater level data in general, but only in this study. However, the authors would like to thank the reviewer for this remark, which shows a lack of references to studies that have been conducted in the past and still are conducted today to evaluate LSMs with different data sets.

The introduction of the manuscript has been modified and the following paragraph has been added (top of page 2 to top of page 3, i.e. lines 35-71):

– Land surface models (LSMs), whether or not coupled to hydrology, have been the subject of numerous studies that have improved them over time and have led to a better description and understanding of the key processes governing exchanges at the interface between the surface and the atmosphere and the surface and the subsurface. These studies have made it possible to evaluate surface models, and even certain parameterizations, by comparing simulation results with different types of observations such as in situ measurements, reanalyses or satellite products. Over time, a number of international measurement campaigns have been organized to evaluate the performance of the models by comparing them with in situ measurements. 
[revised manuscript text omitted]

- We need more information on how the point data were upscaled to allow comparisons with the model grid results. Comparing point data to spatially averaged model results does not make much sense. Furthermore, some grids may contain more stations than others. This will impact the results of the comparison, and needs to be explained better.

- It is not clear what data the reviewer is referring to. In any case, the answer we propose is valid for all types of observations used in the study. First, the model results are not spatially averaged when compared to point data. Rather, the reverse is true, since it is a downscaling from a resolution of a few tens of kilometres to a finer resolution of 8 km for the surface model and up to 1 km for the hydrogeological model. The method is as follows: the SAFRAN analysis is performed on homogeneous zones in terms of horizontal gradients. The analysed fields are then spatially interpolated to a regular 8 km grid taking altitude into account. Thus, the comparison of IR radiation is made between the SAFRAN analysis interpolated at 8km and point data (GLACIOCLIM or LSAF). The horizontal variability of IR radiation at 8 km is small enough to allow a direct comparison with in situ observations. Second, the ISBA model outputs of ground temperature and snow depth profiles are relatively sparse and only a direct comparison between the model outputs and the observations is possible. Finally, with respect to river flows, the MODCOU model grid varies in the range of 8 km to 1 km near the riverbed, and the comparison between the model output and the observed flow is made by considering the flow at the river outlet and the corresponding model grid point in the 1 km hydrological network grid.

The section 3.2 of manuscript was modified and the following paragraph was added at the end, after the description of the different datasets (lines 312-321):

– The SAFRAN analysis is performed on homogeneous zones in terms of horizontal gradients, and the analysed fields are spatially interpolated to a regular 8 km grid taking altitude into account. Thus, the comparison of infrared radiation (IR) is made between the SAFRAN analysis interpolated at 8 km and the local observation. The horizontal variability of IR radiation at 8 km is small enough to allow a direct comparison with in situ observations. Moreover, the ISBA model outputs of ground temperature and snow depth profiles are relatively sparse and only a direct comparison between the model outputs and the observations is possible. Finally, with respect to river flows, the MODCOU model grid varies in the range of 8 km to 1 km near the riverbed, and the comparison between the model output and the observed flow is made by considering the flow at the river outlet and the corresponding model grid point in the 1 km hydrological network grid. This way of locally validating models by comparing the observation to the corresponding model grid point is not new and has been used in many studies in France and elsewhere (Habets et al., 2008; Decharme et al., 2013; Lafaysse et al., 2011; Vergnes et al., 2014).

- Please make it crystal clear how the 470 discharge stations were selected. If there are more stations, why were they not used?

- Only discharge stations with observations for at least half of the days over the total period were kept.

The following sentence was added in section 3.2 (lines 297-298):

- Only gauging stations with observations for at least half of the days over the total period were kept.

- Line 351: some basins are anthropized. I assume this means "urbanized", because agricultural crops are now in the new vegetation classes. If basins are urbanized or semi-urbanized, and the model does not simulate this, then these basins cannot be used to evaluate the model.

- The term anthropized doesn't seem appropriate. It is partly urbanized as highlighted by the reviewer, which means that the presence of urbanization will affect for example the surface runoff by reinforcing the imperviousness. This effect is accounted for in the model by replacing the urban areas by rocks that will facilitate surface runoff. On the other hand, it's true that agricultural crops are part of the vegetation classes and that the model does not represent explicitly the agricultural practices such as irrigation. Moreover, the model does not account for dams whereas some rivers are highly affected by their presence. But we do think that these basins have to be part of the evaluation to identify the weaknesses of the model and put the efforts on developing methods to account for irrigation or the presence of dams.

 "whereas some basins are anthropized." was changed in the manuscript section 4.2 as follows (lines 366-371):

– while some basins are influenced by human activity. In some basins, the human footprint on the landscape is characterized by an increase in urban and agricultural areas and the presence of dams. In the model, urban areas have been replaced by rocks, a type of natural surface, to represent the presence of urban areas that enhance surface runoff. However, the model does not explicitly represent irrigation or the impact of the presence of dams on river flow. The basins impacted by human activity are of great interest for the evaluation as they allow quantifying errors in the system and proposing improvements.

- Line 417: "can probably be attributed to a deficit in the incoming shortwave ration ... or geothermal heating". This seems speculative, it could simply be that the soil or land cover parameters are wrong.

- It is true that there is no evidence that geothermal heating is a source of error in the ground temperature. This was removed from the manuscript and the original sentence in section 4.6:

"As in the previous study of Decharme et al. (2013), a global cold bias (here of about -0.8 K) is observed at each depth, which can probably be attributed to a deficit in the incoming shortwave radiation at the surface and/or to the none representation of the deep earth geothermal heating."

Was changed into:

- "As in the previous study by Decharme et al. (2013), a cold bias (here about -0.8 K) is observed at each depth, which can probably be attributed to a deficit of incoming short-wave radiation at the surface and/or to an incorrect specification of soil physical properties or surface parameters."

- Section 4.7: remote sensing data could have been used to substantiate the results here. At this point, unless I am misunderstanding, no data are used to validate the conclusions.

- It is true that remote sensing data could have been used to consolidate the results. But the choice was first to focus on the evaluation against river discharges, snow depth and soil temperatures, and second to propose a climatological comparison of the Bowen ratio and the evaporation to precipitation ratio of the system before and after the changes. These two objectives are very important for the applications that are used downstream of the system, especially in the departments at MF in charge of hydrology and agriculture.

On the other hand, remote sensing data does not cover the entire period and a fair comparison to the model climatology is not possible.

The beginning of section 4.7 was modified as follows:

- This section compares the climatology of the SIM system before and after the changes made. The aim is to qualitatively identify the impact of the new model on the distribution of energy flows, which is important for certain hydrological or agriculture-related applications.

Some minor issues:
- Add units to the list of numbers at the bottom of page 3.

- The list was changed as follows:

- 0.01 m, 0.04 m, 0.1 m, 0.2 m, 0.4 m, 0.6 m, 0.8 m, 1 m, 1.5 m, 2 m, 3 m, 5 m, 8 m, 12 m

- Top of page 4: be specific, do not state "few tens of centimeters" and "a few meters".

- In section 2.2, the sentence "Heat transfer is resolved over the total depth, while moisture transfer is resolved only over the depth of the roots, which depends on the type of vegetation, a few tens of centimetres for crops and a few metres for forests." was changed into:

- Heat transfer is resolved over the total depth, while moisture transfer is resolved only over the depth of the roots, which depends on the type of vegetation and its geographical location: a maximum of 1.5 m for type C3 crops and 2.5 m for forests in France.

- Line 215-220: this explanation is very unclear.

- In section 2.5, the paragraph "In SAFRAN the analysis is performed on homogeneous areas of several hundreds of square kilometres and an explicit vertical discretization is applied so that the analysis is done every 300 meters. For each grid box $i$, the analysed variables $X^{\cdot}(i)$ are then interpolated on a horizontal 8 km grid, accounting for the averaged elevation of each grid box, and used as input to the ISBA model." was changed into:

- In SAFRAN, the analysis is performed on homogeneous zones of several hundred square kilometres and the vertical component is explicitly considered with to a 300-metre slicing along the vertical. For each grid cell $i$, the analysed variables $X^{a}(i)$ are then interpolated on an 8 km horizontal grid, considering the average altitude of each grid cell. The analysed variables are then used as input to the ISBA surface model.

- Section 4.7: Bowen ration -> Bowen ratio.

- corrected

- There are 17 figures in the paper. This seems like a bit much to me. Can this perhaps be reduced?

- Figure 2 and Figure 7 have been deleted and Figures 16 and 17 have been grouped together.

---

## Author Comment (AC2) · 27 Apr 2020

- The colour black is used for the reviewer questions/remarks.
- The colour green is used to answer questions raised by the reviewer.
- The colour blue is used for the text added to the manuscript.

Summary:

This paper describes the new developments in the Safran-Isba-Modcou (SIM) hydrometeorological model as part of SURFEX v8.0. These new developments cover different aspects of the SIM system: atmospheric forcing, climate fields, land-surface model parameterizations and water budget parameterizations. The authors evaluate the impact of the new developments in an incremental way using different types of observations. They show that the new SIM system, considering all model changes, improves the simulation of daily river discharges and snow depth over a set of catchments and sites covering the French region, compared to the previous SIM system.

The discussion of the results is clear in most of the parts and conclusions are overall justified by the results shown. However, in my opinion the authors do not fully discuss the interactions among the different model changes, therefore explaining the physical mechanisms of some of the results. Also, I found the section on the soil temperature evaluation weak compared to the others. Finally, I have few comments on the introduction and model description parts: these sections are a bit difficult to read and can be improved. All these points are discussed in the main comments below. In summary, I would recommend the acceptance of the paper after major revisions, to make the paper stronger and more attractive to readers.

Main comments

- Introduction and model description: Sect. 1 and Sect. 2 are quite difficult to read. Sentences are not clear in some places, making it difficult to understand the message that the authors want to deliver. I would suggest the authors to improve the readability of these sections. Few examples are reported in the minor comments below.

- The manuscript has been revised to improve the English (request of the first referee), therefore the readability of all the sections of the manuscript has been improved. More comments can be found in the answers to the minor revisions.

- Discussion of the results: The authors performed a set of sensitivity experiments to extract the effect of each model change. However, I have the impression that they do not fully discuss the interactions among the different model changes, and how these relate to the results of the evaluation. For instance, why the SIM_PHY simulation deteriorate the scores in many of the presented metrics (Fig.8, Fig.9, Fig.11)? Is it because of errors in other components, like the atmospheric forcing, which then penalise a more physically complex model? If that is the case, are the other changes reducing such errors, therefore allowing to fully exploit the benefit from the new soil/snow parameterizations? Or are these unrelated? Another example is in the evaluation of the snow depth: the authors propose several hypothesis to explain the improvement of the simulation of the snow depth in SIM_NEW. However, the comparison of all experiments would clearly quantify which processes/changes are responsible for the improvements (see also one of the minor comments below).

- Thank you for the interesting question. In fact, what we see in Fig. 8 for instance is that SIM_REF is simulating the correct modelled to observed river flow (centred around 1) whereas in SIM_PHY this ratio is overestimated. However, in SIM_PHY, as explained in the description of the model, more complexity was added to the model based on a better representation of physics. Moreover, errors in the forcing data show that errors are compensating in SIM_REF. In SIM_PHY, the calculations performed on each of the vegetation type are using the A-gs parameterization of photosynthesis which tends to produce less evaporation over vegetation, leading to more water available in the rivers. On the other hand, it has already been mentioned that the radiation forcing is biased low. The combination of more water available in the soil and less radiative energy to evaporate it leads to an overestimation of the river flows. The more physically based SIM_PHY model is penalised by errors in the forcing. When correcting the IR radiation, SIM_FRC simulation exhibits a large improvement in the scores of river flows, with a modelled to observed ratio peak closer to 1, and a daily efficiency range improved in almost all cases, except perhaps for NSEs lower than 0.4, but the difference to SIM_REF is very small. The implementation of the subgrid topography with the use of elevation bands and of the subgrid hydrology with including a reservoir for snow are essentially impacting hydrology in mountains, and therefore snow and river flows that may be affected by snow melt.

The following paragraph has been added (Section 5.2 of the manuscript):

- The results show that SIM_REF simulates the correct ratio between modelled and observed river flow (centred around 1) whereas in SIM_PHY, this ratio is overestimated. However, in SIM_PHY, as explained in the model description, more complexity has been added to the model based on a better representation of physics. In addition, errors in the forcing data show that errors compensate for each other in SIM_REF, since despite a radiative deficit, river flow is rather well simulated. In SIM_PHY, the calculations performed on each of the vegetation types use the A-gs photosynthesis parameterization, which tends to produce less evaporation on the vegetation, leading to more water available in the rivers. On the other hand, it has already been mentioned that radiative forcing is underestimated. The combination of more water available in the soil and less radiative energy to evaporate it leads to an overestimation of river flows. This is the case of the more physical SIM_PHY model, which is more penalized by errors in forcing. By correcting for IR radiation, the SIM_FRC simulation shows a clear improvement in river flow scores, with a peak of the modelled to observed ratio closer to 1, and an improved daily efficiency range in almost all cases, except perhaps for NSEs below 0.4, but in this case the difference with SIM_REF is very small. The implementation of the subgrid topography with the use of elevation bands (SIM_TOP) and the subgrid hydrology with the inclusion of a snow reservoir (SIM_NEW) essentially impacts the hydrology in the mountains, and thus the snow and river flows that are affected by snowmelt.

- For the evaluation of the snow depth, the comparison can only be made on the 9892 cells, which corresponds to the SIM_REF grid. The snow depth in SIM_FRC, SIM_TOP and SIM_NEW is the same because these experiments use the same IR correction and sub-grid processes related to topography or hydrology are not considered in the evaluation. In terms of snow depth, only SIM_PHY would be different from the other three simulations where the IR correction is applied below 1340m, which limits the interest of such a comparison.

The following paragraph has been added to section 5.3 of the manuscript:

- For the evaluation of the snow depth, the comparison can only be made on the 9892 cells, which corresponds to the SIM_REF grid. The snow depth in SIM_FRC, SIM_TOP and SIM_NEW is the same because these experiments use the same IR correction and sub-grid processes related to topography or hydrology which are not considered in the evaluation. In terms of snow depth, only SIM_PHY would be different from the other three simulations if the IR correction were to be applied below 1340 m, which limits the interest of such a comparison.

- Soil temperature evaluation: I found Sect. 4.6 on the analysis of soil temperature profiles rather weak and with not enough details. The scores of the new system, without a reference, cannot be put into the context of the paper and so are not adding valuable information to the results. A comparison between the different simulations would clarify at least the impact of each change on the bias. The authors state that such biases can be associated to incoming shortwave radiation or lack of geothermal heating, but what about the soil parameterization or soil/surface properties?

- This remark is very relevant, and that is true that the section on the soil temperature results looks too weak. As for the snow depth comparison, the ground temperature in SIM_FRC, SIM_TOP, and SIM_NEW is the same. Only SIM_PHY is different, and exhibits larger biases and rmses at all depths. Such biases can be of course also be associated to soil parameterization and soil and surface properties. It was decided to remove the section 4.6 and to add a discussion on the ground temperature results. Also, the table 3 and the figure showing the ground temperature observation network, as well as the description of the soil observations were removed.

The following paragraph has been added to section 5.1 of the manuscript (lines 471-473):

- We also compared the simulated soil temperatures to the observations made over France. The IR correction on soil temperature has a positive impact and significantly reduces biases and RMSEs (not shown). The results are consistent and of the same order of magnitude as those obtained by Decharme et al (2013).

Minor comments

- Abstract: The main scope of the paper is placed at the very end of the abstract. This could be placed earlier in the text to make clearer the main message of the paper.

- The abstract has been amended and the last sentence has been placed at the beginning:

- This paper describes the impact of the various changes made on the Safran-Isba-Modcou hydrometeorological system (SIM), and demonstrates that the new version of the model performs better than the previous one by making comparisons with observations of daily river flows and snow depths.

- Ln-35–40: This paragraph should be rephrased and clarified. Also references to previously published work on the evaluation of land surface and hydrological models should be introduced in the text.

- The introduction has been revised (following the first referee's remark) and the remark raised here has been dealt with.

- Ln.42: What do you mean by "independent" variable? As the authors stated few lines before, surface energy and water budgets form a coupled system. Please clarify/reformulate.

- The sentence was reformulated (line 72 of the revised manuscript) into:

- In addition, climate models have been evaluated at both global and regional scales through hydrology. Indeed, the coupling between their land surface model and hydrology allows a quantitative assessment to be made, through comparisons to variables such as river flow, groundwater levels and snow depth.

- Ln. 49-50: "... , where modelling contribution of SVAT . . . accounted for in models." this sentence is not clear, please reformulate.

- This paragraph was reformulated (lines 80-83) as:

- Recent initiatives to study the impact of anthropization on water availability, such as those supported by the Global Energy and Water Exchanges (GEWEX) project (Harding et al., 2015), where the contribution of LSMs to modelling appears to be important, show that irrigation needs to be considered in the models (Boone et al., 2019).

- Ln.65-73: The scope of the work can be improved, to make it more precise and easier to read. For instance the authors talk about "new parameterizations" at line 66, but changes to the atmospheric forcing (Ln. 68) are not a parameterization. Also at Ln.69 they talk about "these results", but it is unclear which results are referring to at this point.

- This paragraph was reformulated (lines 97-103) as:

- The objective of this paper is to show how the development of new parameterizations and better atmospheric forcing prescription have improved the performance of the system. The current study, based on numerical simulations covering the period 1958-2018, shows how improvements in atmospheric forcing, land surface model physics and subgrid orography and hydrology improve the modelled river flow and snow depth of the SIM system. It also aims to describe how the model results are affected by each change separately and finally to demonstrate that the new model configuration performs better than the previous one in terms of river flow extremes, and when simulated snow depth or average river flow is compared to observed data.

- Ln. 83: could you be more specific on the horizontal resolution of the SAFRAN analysis?

- The SAFRAN analysis is performed on irregular areas of a few hundred square kilometres. In fact, the size of the area where the analysis is done varies from 400km2 to 1000km2, and then as indicated in the manuscript the horizontal interpolation is done to an 8 km regular grid. In the manuscript a reference to Le Moigne (2002) has been added (line 113) because a description of these areas is given.

- The analysis is carried out over geographical areas covering a few hundred square kilometres (Le Moigne, 2002), and the analysed fields are interpolated to hourly time steps.

- Ln. 83: is the 24h precipitation analysed every 6-hours?

- The analysis of 24h precipitation is performed daily. Manuscript was updated accordingly (line 111-113).

- SAFRAN (Durand et al., 1993; Quintana Segui et al., 2008) performs a 6-hourly analysis of near-surface meteorological variables such as temperature and relative humidity at 2 metres, wind speed, cloud cover and a daily analysis of 24-hour accumulated precipitation.

- Ln. 154: please clarify in which sense you mean "dry", as this can be related to various processes.

- The term "dry" here refers to the soil moisture. The sentence in the manuscript was changed to (line 174):

- However, the Force-Restore scheme is known to be too dry in terms of soil moisture

- Ln. 163-167: this paragraph should be simplified/clarified, to make it clearer that the soil map is not changed between the previous SIM system and the new system (as far as I understood). Ln. 166-167: please clarify that this map was the one used in the "old" version of SIM.

- The beginning of section 2.3 was modified to account for these 2 remarks (lines 182-189):

- In addition to the changes in model physics described above, the land cover and topography databases have been updated to improve the realism of the external parameters of the ISBA model. The hydrogeological database representing the aquifer and the routing network was unchanged. In addition, the soil texture database for France is unchanged. In the former SIM system, the soil texture was based on a soil map provided by the Institut National de Recherches Agronomiques (INRA - King et al., 1995) at a resolution of 1 km. In the new SIM system, texture is defined by the Harmonized World Soil Database (HWSD - Nachtergaele et al., 2012) which is a soil map at 1 km resolution that combines several data sets available worldwide. In particular for France, the INRA soil map mentioned above has been integrated into the HWSD dataset (used in other applications using SURFEX outside France), so this change does not affect the SIM simulations.

- Sect. 2.3: as far as I understood, the main change in the climate fields is the update to ECOCLIMAP2. Hence, I would suggest to expand the discussion of this change, for instance explaining the impacts on the ISBA model? See also next comment.

- The last paragraph of section 2.3 was modified as follows (lines 201-211) to expand the discussion as required and also indicate the use of a new albedo parameterization.

- The impacts of modifying the vegetation fraction input to the ISBA model are multiple and will not be described here in detail (for a detailed comparison, see Faroux et al., 2013). ECOCLIMAP2 has definite advantages, the effects of which are directly reflected in the ISBA model. For example, ECOCLIMAP2 covers a larger time period than the previous version and therefore allows a better representation of the variability of surface parameters. Also, it distinguishes different types of crops that can be modelled separately, and therefore more accurately, with ISBA. The sensors on board satellites have better accuracy and the uncertainty of the measurement is reduced. The vegetation fraction in particular is improved and with it the roughness length of the vegetation which impacts the surface wind by the obstacle effect on near-surface flows. The leaf area index is also improved and its increase leads to a better description of the evaporative fraction, which is key for the energy partitioning in the model. The more realistic surface albedo developed by Carrer et al. (2014) was also used, as Decharme et al. (2013) showed that it improved results at the global scale.

- Ln. 187-193: this sentence is very long. Coud you break it in multiple sentences, better explaining the impact of this change?

- The answer is contained in the respond to the previous remark.

- Sect. 2.4, title: could you specify in the Section title that this is downward infrared radiation?

– Done (line 212).

- 2.4 Evolution of downward infrared radiation

- Ln. 195-198: is the bias related to a problem in the analysis (for instance cloud cover) or a RT model issue?

- The bias is likely due to a problem in the analysis and in the RT model. The cloud cover analysis is computed using T and q profiles from a large-scale atmospheric model that contains biases. The model used to solve the RT is an old model, with a rather low vertical resolution and therefore probably sub-optimal, but which was state-of-the-art in the 1990s.

A sentence has been added to the manuscript (line 218):

- The bias is likely due to a problem in the analysis and in the radiative transfer (RT) model. The cloud cover analysis is computed using temperature and humidity profiles from a large-scale atmospheric model that contains biases. Moreover, the model used to solve the RT is an old model, with a rather low vertical resolution and therefore probably sub-optimal, but which was state-of-the-art in the 1990s.

- Ln. 214: annual mean over which time period?

- Annual mean over the 60-years periods. This was added to the manuscript (end section 2.4, lines 230-231).

- Figure 2 shows the annual average over the 60-year period initial infrared radiation (left panel) and the amount of energy supplied when the correction is applied (right panel).

- Ln. 215-217: could you clarify this sentence? is the analysis done every 300m in the vertical direction?

- This is clarified in the manuscript (lines 233-234):

- In SAFRAN, the analysis is performed on homogeneous zones of several hundred square kilometres and the vertical component is explicitly considered with to a 300-metre slicing along the vertical.

- Ln. 233: why between 3 to 5 layers are necessary, and not more? What it is the vertical discretization between each band? Please clarify.

- The original attempt was to have 10 layers of 300 m from ground to 3000 m. However, this solution appeared to be too expensive and a solution based on the distribution of altitudes in each grid box in 5 classes using deciles q20, q40, q60 and q80 was adopted. The vertical discretization varies from one grid point to another, but is at least equal to 300 m.

The sentence was changed in the manuscript to explain it better (lines 247-248):

- Using a vertical discretization of 300 m at each grid point to represent topographic variability was ideal but too costly. A solution based on the distribution of elevations in each grid cell into five bands represented by the quintiles q20, q40, q60 and q80 was adopted. For each of the 1044 grid points, the vertical discretization varies and is at least equal to 300m. In the end this gives a total of 3878 grid points involved in the calculations of the mountain simulation. Figure 1 (right panel) shows the elevation of the 1044 grid points where the elevation band method is applied.

- Ln. 264: How does the relatively low horizontal resolution of the ERA-40/ERA-I data impact the simulations? The horizontal resolution of ERA-I is ~80km, that is one order of magnitude less than the one used by the SIM grid. I am thinking for instance at regions with a low coverage of surface stations used in the analysis.

- In fact, the SAFRAN analysis does not suffer from a coarse guess as input. SAFRAN tries to be as close as possible to the observations, for temperature, humidity, …, precipitation. The analysis of precipitation even doesn't use large scale information as input. In France, the density of the observation network is very high, because a network dedicated to climatology complements the synoptic network which is less dense. So, there are almost no region with low coverage especially for precipitation which is key for hydrologic purposes.

The manuscript was modified as follows (lines 276-279)

- In France, the density of the observation network is very high, because a network dedicated to climatology completes the less-dense synoptic network. There are therefore practically no regions with poor coverage, especially for precipitation, which is essential for hydrology, and the coarse resolution of the analysis first guess is not an issue.

- Ln. 299-300: were the data cleaned in some way? For instance removing stations with a few number of observations? Or all data have been used to compute the statistics? The latter could introduce some artifact in the statistics. This should be better explained in the text.

- The length of the series is a source of variability in the scores (in particular the number of seasons the stations are open can vary from 1 to 32) but since very few series are complete, it was felt that it was nevertheless more robust to assess the performance of the model to consider as many stations as possible rather than trying to homogenize the length of the series.

The manuscript was modified (lines 308-311) as follows:

- The length of the measurement series and the number of seasons that stations are open are sources of variability in the scores. However, since very few series are complete, the choice was made to evaluate the performance of the model by considering as many stations as possible rather than trying to homogenize the length of the series.

- Ln.318-320: could you specify clearly in the text when the transition from ERA-40 to ERA-I occur in SAFRAN?

- The question raised here pushed the authors to verify in more details which ERA data was used. And it turned out that ERA-40 is used until 2002, then replaced by the operational ECMWF analysis. This means that ERA-I is not used, and the manuscript was corrected (line 275-276) as follows:

- In SAFRAN, the guess of the analysis used is ERA-40 until 2002 and the ECMWF operational analysis thereafter.

- And (line 336-342) as follows:

- In addition to this physical reason, a more technical reason is the change in the large-scale analysis used as boundary conditions to the ERA-40 reanalysis (Uppala et al., 2005), with a priori small changes in the analysed fields. During the production of the ERA-40 reanalysis, the ECMWF operational data assimilation system has evolved considerably and switched to a 4D-var variational method compared to the 3D-var method previously used. This new system has proven to be more accurate and the assimilation of a much larger number of satellite observations has led to a significant improvement in analysis and forecasting, in particular, for the vertical profiles of temperature and relative humidity.

- And (line 457-458) as follows:

- As explained in section 4.1, the calculations of these profiles have varied over time as a result of improvements in the global data assimilation systems used in the ERA-40 reanalysis production.

- And the reference to ERA-Interim was suppressed whereas the reference to ERA-40 was added:

- Uppala SM, Kållberg PW, Simmons AJ, Andrae U, Da Costa Bechtold V, Fiorino M, Gibson JK, Haseler J, Hernandez A, Kelly GA, Li X, Onogi K, Saarinen S, Sokka N, Allan RP, Andersson E, Arpe K, Balmaseda MA, Beljaars ACM, Van De Berg L, Bidlot J, Bormann N, Caires S, Chevallier F, Dethof A, Dragosavac M, Fisher M, Fuentes M, Hagemann S, Hólm E, Hoskins BJ, Isaksen L, Janssen PAEM, Jenne R, McNally AP, Mahfouf JF, Morcrette J-J, Rayner NA, Saunders RW, Simon P, Sterl A, Trenberth KE, Untch A, Vasiljevic D, Viterbo P, Woollen J. 2005. The ERA-40 re-analysis. Q. J. R. Meteorol. Soc. 131: 2961– 3012, 2005.

- Ln. 335-345: What is the reason for the deterioration in the lower part of the CCDF of NSE in SIM_PHY?

- The answer to that question is not straightforward. Most of the stations concerned by a deterioration in the lower part of the CCDF of NSE have a NSE lower than 0.55 and represent approximately 57% of the total number of stations. One part of the explanation comes from the calibration of the subgrid drainage in SIM_REF and not in SIM_PHY as explained in the manuscript. Then, a NSE lower than 0.5 can be considered as a bad simulation. So, both SIM_REF and SIM_PHY have problems in simulating the river flow at those stations. Several reasons can be proposed, and the first one is that some basins are urbanized and this is not well represented in the model. Then as we have seen, there are compensating errors in SIM_REF (correct Qsim to Qobs ratio and too low IR downward radiation).

The manuscript was modified to add this comment (lines 356-359):

- Most of the stations affected by deterioration in the lower part of the NSE CCDF have an NSE below 0.55 and represent about 57% of the total number of stations. Part of the explanation comes from the calibration of the subgrid drainage in SIM_REF which is not done in SIM_PHY.

- Sect.4.3: I would suggest to rename this subsection as it is quite vague at the moment: most of the paper regards the comparison to old SIM.

- The section was renamed (line 387):

- 4.3 Seasonal river flows

- Ln.405: why not adding a third box for SIM_PHY to evaluate the effect of the new snow/soil schemes on the snow depth?

- As already mentioned in the response to the main comments, the comparison to observations and involving SIM_REF can only be made over the 9892 grid boxes and is limited to those below 1340 m (elevation below which IR correction is applied). Therefore, only snow at mid-altitude would be considered and adding a third box with SIM_PHY would not help highlighting the effect of snow and soil schemes on snow depth.

- Ln. 413: "baresoil"–> bare soil

- Section 4.6 has been removed

- Ln.414: At which depth the soil temperature observations are taken? Is any interpolation applied to the data?

- Section 4.6 has been removed (As indicated in the original manuscript, temperatures are measured at 10 cm, 20 cm, 50 cm, and 100 cm, and no interpolation was applied)

- Ln.440-449: It would be nice to explicitly link this discussion on the changes of Evap/Precip with the changes in the discharge mean bias.

- A sentence has been added to make this link (lines 447-449):

- In SIM_NEW, the ratio of simulated to observed flow is in excess whereas it is better simulated in SIM_REF with a peak centred around 1. This result is consistent with an evaporation deficit in SIM_NEW compared to SIM_REF.

- And the last sentence of the paragraph was changed to (lines 450-452):

- This result shows that the sensible heat flux in SIM_NEW is much higher than in SIM_PHY, mainly due to the increased incoming infrared radiation, which partially compensates for the evaporation deficit.

- Ln. 489: I would rephrase this sentence for readers not familiar with detailed snow models.

- Sentence rephrased to (lines 510-512):

- At the same time, as described in section 2, the snow model has been revised to improve vertical layering, snow compaction and solar energy transmission within the snowpack through the use of spectral albedo, as is done in more advanced models.

Comments on the figures

- Generally, the figure captions should be improved to make them more self-explanatory.

- The figure's captions were improved

-

Figure 1: Height of the topography of the 9892 cells of the SIM grid (left) and the 3878 cells of the mountain SIM grid (right). The cells of the mountain grid correspond to the 1044 points having an altitude greater than 500 m and described vertically by several layers. Zones in yellow correspond to the Seine and Rhone aquifers. The dotted line delimits the Alps mountain.

Figure 2: Annual average of uncorrected (left) and corrected (right) downward longwave infrared radiation from SAFRAN analysis.

Figure 3: Maps of annual average of the SAFRAN analysis for the period 1958-2018 of (a) air temperature at 2 meters, (b) specific air humidity at 2 meters, (c) wind speed at 10 meters, (d) total annual precipitation, (e) direct solar radiation, and (f) diffuse solar radiation.

Figure 4: Annual average of the SAFRAN analysis of (a) air temperature at 2 meters, (b) specific air humidity at 2 meters, (c) wind speed at 10 meters, (d) direct solar radiation, (e) diffuse solar radiation, (f) infrared radiation, and (e) total precipitation rate.

Figure 5: Comparison of the NSE CCDF (left panel) and the simulated to observed flow ratio (right panel) for SIM_REF (dashed blue line), SIM_PHY (solid red line), SIM_FRC (solid cyan line), SIM_TOP (solid green line), and SIM_NEW (solid orange line).

Figure 6: Maps of the difference in mean NSE for NSE>0 between simulations: (a) SIM_PHY and SIM_REF, (b) SIM_FRC and SIM-PHY, (c) SIM_TOP and SIM_FRC, (d) SIM_NEW and SIM_TOP.

Figure 7: Map of the difference in mean NSE for NSE>0 between SIM_NEW and SIM_REF (left panel), and SIM_NEW NSE map (right panel).

Figure 8: Taylor diagrams of seasonal river flows for the different experiments over the period 1958-2018.

Figure 9: Taylor diagram of Q10 and Q90 deciles of river flows over the period 1958-2018.

Figure 10: Comparison of monthly river flows with SIM_NEW for the Garonne at Lamagistère over the period 1958-2018.

Figure 12: Maps of mean annual Bowen ratio (a) and evaporation to precipitation ratio (b) for SIM_NEW on average over period 1958-2018.

Figure 13: Mean annual evaporation to precipitation ratio (a), and Bowen ratio (b), for experiments SIM_REF, SIM_PHY, and SIM_NEW.

- In the caption of Figure 1, the definition of a "mountain grid cell" should be added.

- Done

- Some of the figures could be merged together, for instance Fig. 7 with Fig. 8 and Fig. 9 with Fig. 10, for conciseness.

- As compared to the initial manuscript, some figures have been removed (2, 6, 7) and 16 and 17 have been merged together.

---

## Referee Report (RR1)

**Reviewer's comment on manuscript gmd-2020-31**

**Summary:**

The authors have improved the original manuscript, in particular in the introduction and model's description sections. They also addressed my main points, with the inclusion of a discussion regarding the role of compensating errors among the different simulations. However, I still have some comments on specific points of the revised manuscript, which I would like the authors to consider. For this reason, I recommend the acceptance of the manuscript after minor revisions.

**Specific comments**

Page 2 Ln 36: "coupled to hydrology" is not clear, I would suggest "... coupled to hydrological models"

Page 2 Ln 38-41: These two sentences could be improved, as it sounds a bit of a repetition.

Page 2 Ln 42: Atmospheric biases would be introduced in land surface model simulations even in offline simulations, for instance a precipitation bias in the atmospheric model leads to a spurious anomaly in soil moisture. Could you please clarify?

Page 5 Ln 145: "as realistically as possible" seems a too strong statement. Boone and Etchevers' (2001) 3-layer snow scheme is an intermediate complexity snow scheme, that is in between simple bulk models and very detailed snow schemes used for avalanche forecasting.

Page 8 Ln 249-250: I would suggest to put the range of values used in the vertical discretization in the text, in order to be clearer.

Page 10 Ln 309-311: Whilst I understand that all data were used, it is important to highlight if any stations contained large periods of missing data during the analysis period, and if so how this influences the conclusions drawn.

Page 11 Ln 336-341: As far as I understand, reanalysis use a fixed model and data assimilation system, for consistency, throughout the production. So I am not sure how these sentence relate to the behaviour in the forcing fields discussed in this subsection. Could the authors clarify this?

Page 14 Ln 420-421: The authors suggest that the improvement in altitude resolution, I think due to the new orography map implemented, can be responsible for the improvements. However, at Page 6, Ln 94-96 they state that "the impact of using SRTM90 is rather limited". Could the authors clarify the link between these two statements?

Page 14 Ln 421-423: Is this a hypothesis to explain the improvement in SIM_NEW? Or is it a diagnostic technique that could be used to further improve the results? Can the authors clarify this sentence?

Page 14 Ln 423-424: I would say that also the new vegetation maps can play a role in the improvements.

Page 15, Sect. 5.2: Many statements in this subsection could be improved for clarity.

Page 16, Ln 482: "...less radiative energy to evaporate  leads to an overestimation of river flows" (remove "it")

Page 16 Ln 492-494: This sentence needs more clarity, as it is currently difficult to follow.

Page 16 Ln 491-494: So the improvements in snow depth could be mainly attributed to the improvements in the snow/soil physics and the new climate fields, is it right? It would be interesting to know which one (new physics or new land cover fields) play the larger role.

Page 17 Ln 512: "as **it** is done in…" (add "it")

Page 18 Ln 555: the word "Formula" be more appropriate than "Principle".

---

## Author Response (AR2)

[revised manuscript text omitted]

**Reviewer's comment on manuscript gmd-2020-31**

**940** **Summary:**

The authors have improved the original manuscript, in particular in the introduction and model's description sections. They also addressed my main points, with the inclusion of a discussion regarding the role of compensating errors among the different simulations. However, I still have some comments on specific points of the revised manuscript, which I would like the authors to consider. For this reason, I recommend the acceptance of the manuscript after minor revisions.

**945** **Specific comments**

Page 2 Ln 36: "coupled to hydrology" is not clear, I would suggest "... coupled to hydrological models"

"coupled to hydrology" was changed into "coupled to hydrological models"

Page 2 Ln 38-41: These two sentences could be improved, as it sounds a bit of a repetition.

The two sentences have been grouped into a single one:

**950** "These studies, which include international measurement campaigns or more regional or even local initiatives, have made it possible to evaluate surface models, and even certain parameterizations, by comparing simulation results with different types of observations such as in situ measurements, reanalyses or satellite products."

Page 2 Ln 42: Atmospheric biases would be introduced in land surface model simulations even in offline simulations, for instance a precipitation bias in the atmospheric model leads to a spurious anomaly in soil moisture. Could you please clarify?

**955** It is true that offline simulations also suffer from atmospheric biases. However, offline simulations allow a better control of these biases through the use of observations or reanalysis. When coupled with the atmosphere, LSMs are also impacted by the intrinsic biases of the atmospheric model, without the possibility of correcting them (except by using data assimilation, etc.).

The manuscript was modified and the sentence "Simulations were carried out offline, i.e. decoupled from the atmosphere, to **960** avoid introducing atmospheric biases into the surface schemes." was changed into "Simulations were carried out offline, i.e. decoupled from the atmosphere, to limit the impact of potential atmospheric biases in the surface schemes by constraining atmospheric forcing through observations, when possible."

Page 5 Ln 145: "as realistically as possible" seems a too strong statement. Boone and Etchevers' (2001) 3-layer snow **965** scheme is an intermediate complexity snow scheme, that is in between simple bulk models and very detailed snow schemes used for avalanche forecasting.

A more moderate sentence replaces the original: "as realistically as possible" was changed into "realistically with a simple model"

Page 8 Ln 249-250: I would suggest to put the range of values used in the vertical discretization in the text, in order to be **970** clearer.

"For each of the 1044 grid points, the vertical discretization varies and is at least equal to 300m"

was changed into

"For each of the 1044 grid points, the vertical discretization varies spatially, and the vertical discretization is variable when the maximum altitude difference in the grid exceeds 300m. Thus, for all of the 1044 points, the minimum difference (23m) between two consecutive bands is obtained in medium mountains, for altitudes of 385m, 694m, 717m, while the maximum difference (986m) is obtained in high mountains for altitudes of 525m, 861m, 1847m, 2013m."

Page 10 Ln 309-311: Whilst I understand that all data were used, it is important to highlight if any stations contained large periods of missing data during the analysis period, and if so how this influences the conclusions drawn.

This remark was already addressed in the first review, and the following answer was given: "The length of the measurement series and the number of seasons that stations are open are sources of variability in the scores." Difficult to say more without a complete statistical study which is not foreseen here.

Page 11 Ln 336-341: As far as I understand, reanalysis use a fixed model and data assimilation system, for consistency, throughout the production. So I am not sure how these sentence relate to the behaviour in the forcing fields discussed in this subsection. Could the authors clarify this?

It is perfectly correct that reanalyses use a fixed version of the model to ensure consistency throughout production. However, during production, the reanalysis is influenced by changes in the density of observations that are assimilated and also potentially by certain modifications to the system. As explained by Uppala for the ERA-40, there was a substantial change in the number of observations assimilated from 1979 to 2002 and the operational use of the 4DVar also had an impact on the ERA-40 reanalysis due to changes in the calculation of covariance errors in observations and guess (and other more technical changes that I will not mention here). In order to clarify the sentence in question, the manuscript has been amended and the original sentence has been modified:

"In addition to this physical reason, a more technical reason is the change in the large-scale analysis used as boundary conditions to the ERA-40 reanalysis (Uppala et al., 2005), with a priori small changes in the analysed fields. During the production of the ERA-40 reanalysis, the ECMWF operational data assimilation system has evolved considerably and switched to a 4D-var variational method compared to the 3D-var method previously used. This new system has proven to be more accurate and the assimilation of a much larger number of satellite observations has led to a significant improvement in analysis and forecasting, in particular, for the vertical profiles of temperature and relative humidity."

Was changed into (line 334):

"In addition to this physical reason, more technical reasons such as changes over time in the density of assimilated observations or changes in the ECMWF operational system may have affected the ERA-40 reanalysis. Although the model used in the reanalysis is a frozen version, the reanalysis system includes input observations whose density varies significantly over time (Uppala et al., 2005). In addition, during the production of the ERA-40 reanalysis, the ECMWF operational data assimilation system has evolved considerably and switched to a 4D-var variational method (1997) compared to the 3D-var method previously used. As a consequence, the calculation of the error covariances of the observations and the guess were revised in the 4D-var but also 3D-var and impacted directly the ERA-40 reanalysis. "

Page 14 Ln 420-421: The authors suggest that the improvement in altitude resolution, I think due to the new orography map implemented, can be responsible for the improvements. However, at Page 6, Ln 94-96 they state that "the impact of using SRTM90 is rather limited". Could the authors clarify the link between these two statements?

Page 6 line 94-96, it is stated that the impact of the SRTM90 is rather limited. This is true because we are comparing an average orography at a resolution of 8 km and at this resolution, GTOPO30 or SRTM90 give quite good results and above all quite close. Page 14, line 466, the main impact comes from the sub-grid orography. However, the finer resolution of SRTM90 allows better vertical discretization of sub-grid orography.

The sentence (line 420-421) "These results are consistent with the improvement in altitude resolution in mountainous areas, which reduces the altitude differences between the simulated grid cells and the observation stations. "

was changed into:

"These results are consistent with improved altitudinal discretization in mountainous areas, which reduces the altitude differences between the simulated grid cells and the observation stations."

and the sentence (line 466) "due to the higher resolution orography and the representation of sub-grid orography in the mountains"

was changed into:

"due to the representation of sub-grid orography in the mountains, enhanced by a higher resolution of the orography which allows for finer vertical discretization"

Page 14 Ln 421-423: Is this a hypothesis to explain the improvement in SIM_NEW? Or is it a diagnostic technique that could be used to further improve the results? Can the authors clarify this sentence?

The sentence referred to is "Slight improvements in SIM_NEW scores can also be obtained by linear interpolation at station altitudes of the simulated snow heights at the two closest altitude bands." This is not a hypothesis to explain the improvement in SIM_NEW. This is to emphasize that the simulated snow heights at the two layers surrounding the observation could be used to calculate an average of the snow height at the height of the observation. However, the nearest point was chosen to use the same treatment as in SIM_REF.

The text was changed into:

"Slight improvements in SIM_NEW scores can also be obtained by interpolating linearly the simulated snow depths at the two layers surrounding the observation. However, the point closest vertically to the observation was chosen, in order to use the same selection as in SIM_REF."

Page 14 Ln 423-424: I would say that also the new vegetation maps can play a role in the improvements.

The sentence "It should also be noted that improvements in the snow parameterization should also explain some of the improvement in scores (Decharme et al., 2016). "

was changed into:

"It should also be noted that improvements in the snow parameterization should also explain some of the improvement in scores (Decharme et al., 2016), but also the use of more accurate vegetation maps."

Page 15, Sect. 5.2: Many statements in this subsection could be improved for clarity.

The subsection was revised to improve its clarity (?). The discussion on SIM_REF and the other model configurations have been more clearly separated. Then a few modifications were also done:

The 3 first sentences: "The results show that SIM_REF simulates the correct ratio between modelled and observed river flow (centred around 1) whereas in SIM_PHY, this ratio is overestimated. However, in SIM_PHY, as explained in the model description, more complexity has been added to the model based on a better representation of physics. In addition, errors in the forcing data show that errors compensate for each other in SIM_REF, since despite a radiative deficit, river flow is rather well simulated. In SIM_PHY, the calculations performed on each of the vegetation types".

were changed into:

"The results show that SIM_REF simulates the correct ratio between modelled and observed river flow (centred around 1) whereas in SIM_PHY, this ratio indicates an overestimation. However, errors in the forcing data show that errors compensate for each other in SIM_REF, since despite a radiative deficit, river flow is rather well simulated. In SIM_PHY, as explained in the model description, more complexity has been added to the model based on a better representation of physics. The calculations, performed on each of the vegetation types,"

The sentence "This is the case of the more physical SIM_PHY model, which is more penalized by errors in forcing." was deleted because it didn't bring more information.

Page 16, Ln 82: "...less radiative energy to evaporate **it** leads to an overestimation of river flows" (remove "it")

Done (line 505 of new manuscript)

Page 16 Ln 492-494: This sentence needs more clarity, as it is currently difficult to follow.

For the evaluation of the snow depth, the comparison can only be made on the 9892 cells, which corresponds to the SIM_REF grid. The snow depth in SIM_FRC, SIM_TOP and SIM_NEW is the same because these experiments use the same IR correction and sub-grid processes related to topography or hydrology which are not considered in the evaluation

was changed into (5.3):

For the evaluation of the snow depth, the comparison can only be made on the 9892 cells, which corresponds to the SIM_REF grid. In addition, in order not to disadvantage SIM_REF and to assess the impact of changes in physics and atmospheric fields, sub-grid processes in SIM_TOP and SIM_NEW were not considered in the evaluation (the additional vertical levels of the 1044 cells were not used). Thus, the snow depth simulated in SIM_FRC, SIM_TOP and SIM_NEW is the same because all these experiments use the same correction for infrared radiation.

Page 16 Ln 491-494: So the improvements in snow depth could be mainly attributed to the improvements in the snow/soil physics and the new climate fields, is it right? It would be interesting to know which one (new physics or new land cover fields) play the larger role.

Yes, the impact is related to snow and soil physics, new physiographic fields and land cover, and new atmospheric forcing. It would be interesting to test the impact of these different components separately through a specific study. Decharme et al (2016) have already evaluated the impact of the improved snow scheme in ISBA, as well as the effect of radiative forcing.

075     However, the main objective of the snow evaluation was to obtain a global comparison between the old SIM_REF version and the new SIM_NEW version.

Page 17 Ln 12: "as **it** is done in..." (add "it")

Done (line 536)

080     Page 18 Ln 55: the word "Formula" be more appropriate than "Principle".

Done (line 579)

---

## Author Response (AR3)

[revised manuscript text omitted]

**Topical Editor's comment on manuscript gmd-2020-31**

920 • P.6, l.201; To better answer the referee's remark, maybe add "in practice" in ""Note that the impact of using SRTM90 is rather limited".

The sentence has been changed into (P.6, L.193):

925 "Note that in practice the impact of using SRTM90 is rather limited."

• P.8, l.257: I don't fully understand the sentence "For each of the 1044 grid points, the vertical discretization varies spatially, and the vertical discretization is variable when the maximum altitude difference in the grid exceeds 300m. Thus, for all of the 1044 points, the minimum difference (23m)
930 between two consecutive bands is obtained in medium mountains, for altitudes of 385m, 694m, 717m, while the maximum difference (986m) is obtained in high mountains for altitudes of 525m, 861m, 1847m, 2013m. In the end this gives a total of 3878 grid points involved in the calculations of the mountain simulation." Could you please provide a clearer description providing simply the minimum and maximum mesh vertical extent?
935

The horizontal resolution varies at each grid point and the vertical resolution varies irregularly with strong spatial variability. The sentence was changed into (P.8, L.249-250):

"For each of the 1044 grid points, the vertical discretization varies spatially, and the irregular vertical
940 mesh ranges from 23m in medium mountains to 986m in high mountains."

• P.14, l.447: I propose to change "can be" by "could have been"?

945 The sentence has been changed into (P.14, L.432):

"Slight improvements in SIM_NEW scores could have been obtained by interpolating linearly the simulated snow depths at the two layers surrounding the observation. However, the point closest vertically to the observation was chosen, in order to use the same selection as in SIM_REF."
950
• P.14, l.450: I propose to change "It should also be noted that improvements in the snow parameterization should also explain some of the improvement in scores (Decharme et al., 2016), but also the use of more accurate vegetation maps." with "It should also be noted that improvements in the snow parameterization, but also the use of more accurate vegetation maps, can explain some of the
955 improvement in scores (Decharme et al., 2016)."

The sentence has been modified as suggested by the editor (P.14, L.434-435).

• P.16, l.507-508: I propose to change "...errors in the forcing data show that errors compensate ..." for
"However, good results in SIM_REF are due to error compensation since ..."

The sentence has been modified as suggested by the editor (P.15, L.490-491).

• Paragraph 5.3 on the snow depth: can you clarify what the final conclusion is?
Yes, in fact SIM_FRC, SIM_TOP and SIM_NEW have the same snow simulation because the sub-grid
parameterizations are not activated and the IR correction is the same on the 9892 grids that make up the
SIM_REF and SIM_PHY grid. Only SIM_PHY is different below 1340m because the IR forcing is not
corrected (no subgrid). So, in these conditions the 4 simulations are almost equivalent on the 9892 grids.
But if we consider the 1044 more meshes then the snow is better simulated than in SIM_PHY. The
choice is not to consider these meshes in order to make a fair comparison with SIM_REF.
So, it's true that if we limit ourselves to 9892 meshes then the SIM_REF SIM_PHY comparison is
enough. Snow is better simulated in SIM_NEW on the 1044 meshes but we don't use it in the
comparison with SIM_REF (it is used to improve water flows).

Paragraph 5.3 has been changed into (P.16, L.505-513):

The snow depth simulation is of equivalent quality on the 9892 meshes in SIM_FRC, SIM_TOP and
SIM_NEW because the same IR correction is applied. On the other hand, the subgrid representation of
the topography improves the realism of SIM_TOP and SIM_NEW in terms of snow depth but applies
only to the 1044 additional grid cells. However, for the evaluation of the snow depth, the comparison
can only be made on the 9892 cells, which corresponds to the SIM_REF grid. In addition, in order not
to disadvantage SIM_REF and to assess the impact of changes in physics and atmospheric fields, sub-
grid processes in SIM_TOP and SIM_NEW were not considered in the evaluation (the additional
vertical levels of the 1044 cells were not used). It was decided to present the fairest comparison with
SIM_REF so only considering SIM_PHY. Under these conditions where subgrid effects are not
activated, SIM_PHY is quite close to the other three simulations, the only difference is related to the
change in IR forcing, limited below 1340m.